# Electrification at water–hydrophobe interfaces

Jamilya Nauruzbayeva[1,3], Zhonghao Sun [2,3], Adair Gallo Jr.[1,3], Mahmoud Ibrahim[1], J. Carlos Santamarina [2] & Himanshu Mishra [1✉]

The mechanisms leading to the electrification of water when it comes in contact with hydrophobic surfaces remains a research frontier in chemical science. A clear understanding of these mechanisms could, for instance, aid the rational design of triboelectric generators and micro- and nano-fluidic devices. Here, we investigate the origins of the excess positive charges incurred on water droplets that are dispensed from capillaries made of polypropylene, perfluorodecyltrichlorosilane-coated glass, and polytetrafluoroethylene. Results demonstrate that the magnitude and sign of electrical charges vary depending on: the hydrophobicity/hydrophilicity of the capillary; the presence/absence of a water reservoir inside the capillary; the chemical and physical properties of aqueous solutions such as pH, ionic strength, dielectric constant and dissolved $CO_2$ content; and environmental conditions such as relative humidity. Based on these results, we deduce that common hydrophobic materials possess surface-bound negative charge. Thus, when these surfaces are submerged in water, hydrated cations form an electrical double layer. Furthermore, we demonstrate that the primary role of hydrophobicity is to facilitate water-substrate separation without leaving a significant amount of liquid behind. These results advance the fundamental understanding of water-hydrophobe interfaces and should translate into superior materials and technologies for energy transduction, electrowetting, and separation processes, among others.

[1] King Abdullah University of Science and Technology (KAUST), Water Desalination and Reuse Center (WDRC), Division of Biological and Environmental Sciences and Engineering, Thuwal 23955 - 6900, Saudi Arabia. [2] King Abdullah University of Science and Technology, Ali I. Al-Naimi Petroleum Engineering Research Center (ANPERC), Division of Physical Science and Engineering, Thuwal 23955 - 6900, Saudi Arabia. [3]These authors contributed equally: Jamilya Nauruzbayeva, Zhonghao Sun, Adair Gallo Jr. ✉email: himanshu.mishra@kaust.edu.sa

Water becomes electrified when it comes in contact with hydrophobic media. Electrification underlies various curious phenomena, such as the electrostatic manipulation of droplets placed on hydrophobic surfaces[1–3] and Kelvin generators[4–6]. The electrification of water against hydrophobic surfaces (hereafter referred to as water-hydrophobe interfaces) plays an important role in various applied and natural contexts, such as pipetting[7–9], triboelectric power generation[10–12], hydrogen generation[13,14], mitigating dust deposition on solar panels[15], preventing fire hazards in granular flows[16], and precipitation and thundercloud charging[17,18]. However, the causes and mechanisms underlying this electrification process are still intensely debated[11,14,19–37].

A variety of mechanisms have been put forth to explain electrification of water in contact with solid/liquid/gaseous hydrophobes, including the specific adsorption of hydroxide ions[38–48] and hydronium ions[14,49–55], the dipolar organization of interfacial water[19,22,56], the partial charge transfer between the O and H atoms of interfacial water[57,58] or between interfacial water and oil molecules[59], the adsorption of bicarbonate ions due to the dissolution of ambient $CO_2$[60], contamination[61–66], reactive chemical groups[8,9,67–69], electrons trapped on the surface of insulators[70–72] and mechanoradicals[73,74]. With the exception of surface-bound electrons, these mechanisms assume that common hydrophobic surfaces such as polypropylene and perfluorocarbons are electrically neutral in air. In this work, we designed elemental laboratory experiments to answer the following interrelated questions:

i. Why do water-hydrophobe interfaces become electrically charged?

ii. How do the properties of aqueous solutions, solid surfaces and the environment impact the electrification of water at water-hydrophobe interfaces?

iii. Could other liquids besides water become electrified when brought into contact with hydrophobic surfaces? What is the role of hydrophobicity in the context of electrification at water-hydrophobe interfaces?

Based on this experimental investigation, we deduce that the surfaces of common hydrophobes, such as polypropylene, FDTS, and PTFE, are negatively charged. Thus, when these surfaces come into contact with a liquid containing solvated ions, such as water, cations form an electrical double layer at the interface in accordance to the electrical double layer theory[75].

## Results

**Experimental setup**. We quantitatively investigated the electrification of deionized water droplets dispensed from polypropylene pipette tips and borosilicate glass capillaries (see Methods). Glass capillaries allowed us to precisely control the solid-liquid interfacial tension. For instance, freshly cleaned glass capillaries were superhydrophilic, characterized by ultralow apparent contact angles, $\theta_r \approx 5°$. To render them hydrophobic, we covalently grafted perfluorodecyltrichlorosilane (FDTS) onto them through a molecular vapor deposition technique (Methods). We characterized wetting by measuring advancing ($\theta_A$) and receding ($\theta_R$) contact angles using sessile deionized water droplets of volume $\approx 2$ μL dispensed/retracted at $0.2$ μL s$^{-1}$, and found them to be $\theta_A = 105° \pm 1°$ and $\theta_R = 72° \pm 1°$ respectively (Methods). The polypropylene surfaces of the pipette tips exhibited $\theta_A = 113° \pm 2°$ and $\theta_R = 62° \pm 2°$. Hereafter, deionized water is referred to as water. Supplementary Fig. 1 presents AFM scans of the FDTS-coated glass and polypropylene surfaces.

Two complementary experimental techniques were deployed to investigate the electrification of water. The first technique used pendant droplets of controlled volume (10–20 μL) and surface area, formed at the tip of hydrophobic and hydrophilic capillaries. We recorded the formed droplets' behavior inside a parallel plate capacitor, which comprised of two $100 \times 100$ mm$^2$ aluminum plates that ensured a uniform electric field in the central region of $<4 \times 4 \times 4$ mm$^3$ occupied by the droplets (Fig. 1a, b, Supplementary Section 2, and Supplementary Fig. 2). Two scenarios were tested: (i) the capillary was fully filled with water before producing the droplet at the end of the capillary inside the capacitor (Fig. 1a); and (ii) the capillary was filled with air when the droplet was formed, akin to standard pipetting (Fig. 1b).

The second technique deployed to quantify the electrification of water used an ultrasensitive electrometer (with a detection limit of 10 fC) equipped with a Faraday cup made of aluminum sheet to shield external interferences[76] (Fig. 1c, Supplementary Fig. 3, Methods). Electrical charges on pendant droplets were measured by dispensing them into the Faraday cup. Together, these two techniques enabled us to investigate the effects of surface wettability (hydrophilicity and hydrophobicity) and liquid properties (ionic strengths, pH, and dielectric constants) on the electrification of water.

**Pendant droplets under uniform electric fields**. We used high-speed imaging to quantify the excess charges ($q$) carried by the droplets through their deflections under uniform electric fields. The balance of the electrostatic ($F_E$) and gravitational forces ($F_G$) acting on the pendant droplets gave rise to tilting angles, α, (Fig. 2a-b) as a function of the applied voltage ($V$) as:

$$\tan \alpha = F_E / F_G \tag{1}$$

$$F_G = mg \tag{2}$$

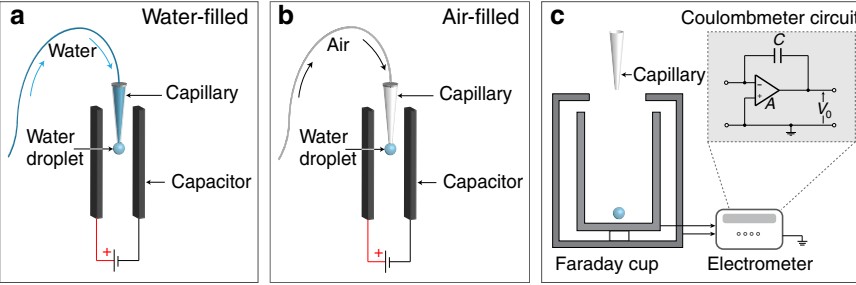

**Fig. 1 Schematics of the experimental set-ups. a** Droplets formed at the tip of a water-filled capillary (hydrophobic or hydrophilic) inside a parallel plate capacitor. **b** Droplets formed at the tip of an air-filled capillary. **c** Direct measurement of electrical charges carried by water droplets using a Faraday cup (comprising aluminum sheet) connected to an electrometer. The inset demonstrates the circuit of the electrometer, where $C$, $V_0$, and $A$ refer to the capacitance, the voltage across the capacitor, and a low current input amplifier, respectively[76].

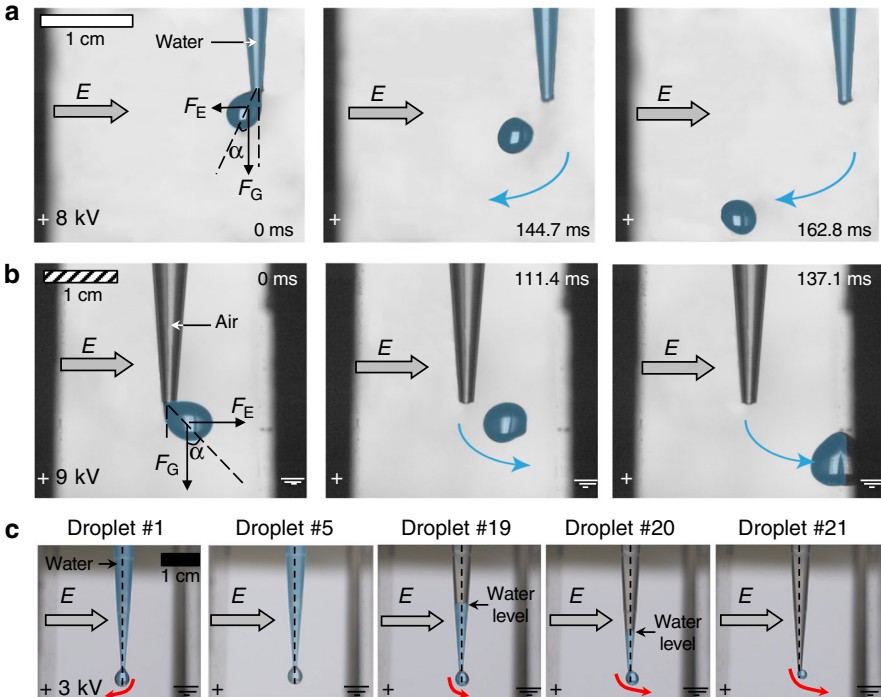

**Fig. 2 Pendant water droplets under uniform electric fields. a** When the capillary is water-filled, the pendant droplets are deflected towards the positively charged plate (we emphasize water using false blue color). The direction of the electric field, $E$, is shown by the gray arrows; free-body diagrams represent the forces due to gravity ($F_G$) and electrostatics ($F_E$) acting on the water droplets. **b** When the capillary is air-filled, the droplets are repelled by the positively charged plate. **c** Photographs of sequential 10 μL water droplets dispensed from a partially filled polypropylene capillary (initial volume: 200 μL) under a uniform electric field. The red arrows point to the directions of the droplets' deflections and the dashed vertical lines are drawn to assist in the visualization.

$$F_{E} = Eq = \frac{V}{L}q \tag{3}$$

where $m$ is the mass of the drop, $g$ is the acceleration due to gravity, $L$ is the distance between the plates of the capacitor, and $E$ is the uniform electric field inside the capacitor ($E = V/L$). Then, the excess charge carried by the droplets ($q$) is:

$$q = \frac{mg \tan \alpha}{E} \tag{4}$$

and the surface charge density, defined as the number of electronic charges (#) per μm², is:

$$\sigma = q/A_o, \tag{5}$$

where $A_o$ is the solid-liquid interfacial area occupied by the water droplet inside the pipette prior to dispensing.

The behavior of the pendant water droplets formed below the hydrophobic polypropylene and FDTS-coated capillaries under uniform electric fields varied dramatically depending on whether the capillary above the droplet was water-filled or air-filled. When the capillary was water-filled, the droplets were deflected towards the positively charged plate of the capacitor (Fig. 2a, and Supplementary Movie 1), whereas when the capillary was air-filled, the droplets were repelled away from the positively charged plate (Figs. 2b and 3, and Supplementary Movie 2). The tilting angles increased with the electric field strength, eventually leading to the detachment of the droplet from the capillary above; the trajectories followed straight lines along the vector sum of the forces due to gravity and electrostatics (Supplementary Movies 1 and 2). In the air-filled scenario, the estimated charge densities of droplets dispensed from hydrophobic and hydrophilic capillaries remained unaffected by the electric fields (Fig. 3). On the other hand, in the water-filled scenario, the charge densities scaled with the electric field strengths for both hydrophobic and hydrophilic capillaries (Fig. 3 and the explanation is presented in the Discussion section).

We expanded the experiment to investigate the transitional behavior of the droplets as a function of the amount of water in the capillary above. Specifically, we filled the polypropylene capillaries with 200 μL of water, placed them inside a uniform electric field, and systematically dispensed ~10 μL droplets from them, while recording the behavior of each droplet. The first four droplets were attracted to the positive plate (similarly to the case described above for the water-filled capillary) the next two droplets did not show any deflection, and all the subsequent droplets were repelled away from the positive plate. Clearly, the droplets became increasingly repelled by the positive plate as the capillary emptied (Fig. 2c, Supplementary Movie 3).

In contrast to the hydrophobic capillaries, when hydrophilic glass capillaries were air-filled, the pendant water droplets were attracted to the positive plate, albeit only very mildly, and their estimated charge density did not scale with the electric field (Fig. 3). However, when these capillaries were water-filled, the droplets were attracted to the positive plate and the charge density scaled with the electric field (Fig. 3). The latter behavior was similar to that of water droplets dispensed from water-filled hydrophobic capillaries, except in this case the charge density was lower and high electric fields were required to detach the droplets from the capillaries.

We repeated these experiments using methanol and hexadecane (Fig. 3). The pendant droplets were not deflected under uniform electric fields as high as ~2000 V cm⁻¹ regardless of the hydrophobicity/hydrophilicity of the capillary. While higher electric fields may eventually induce deflection[20], but such studies are beyond the scope of this work as we are focused on aqueous interfaces specifically.

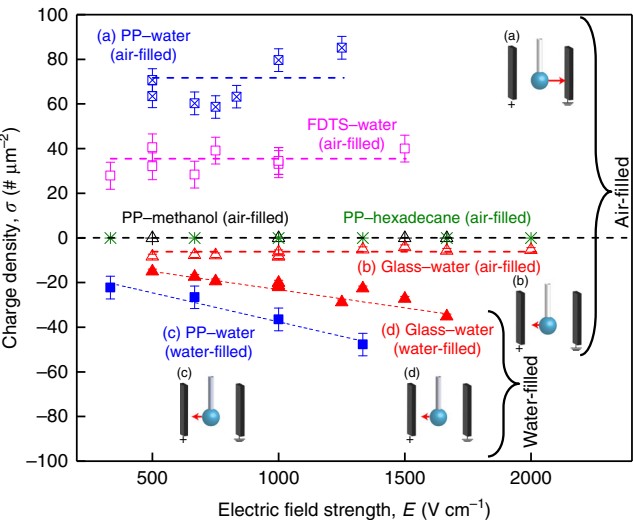

**Fig. 3 Experiments inside a parallel plate capacitor.** Correlations between the charge density of pendant droplets and electric field strengths for water-filled and air-filled hydrophobic and hydrophilic capillaries (based on Supplementary Movies 4–10). Insets depict experimental configuration with the pendant water droplets inside the parallel plate capacitor with the positively charged plate on the left-side. Note: the red arrows present qualitatively the direction and the relative magnitudes of the deflections experienced by water droplets under various scenarios. When the hydrophobic capillaries were **a** air-filled (or **c** water-filled), the pendant water droplets repelled away from (or attracted towards) the positively charged plate. In the case of the hydrophilic glass capillaries, the droplets always tilted towards the positive plate of the capacitor, and we detected very low charges in **b** the air-filled case, and higher charges in **d** the water-filled case. Note: we found no deflection for droplets of methanol and hexadecane dispensed from air-filled hydrophobic capillaries in the specified range of electric fields. We obtained the charge density along the y-axis by normalizing the charge of the droplet obtained from the force balance [Eqs. (1–5)] and liquid-solid interfacial areas of the dispensed droplets. (Dotted lines have been added to guide the eye.) Error bars represent the standard deviation of duplicates. The hollow symbols represent the air-filled cases, and the solid symbols represent the water-filled cases for each interface (Color scheme: blue: PP-water; red: glass-water; magenta: FDTS-water; black: PP-Methanol; green: PP-hexadecane).

**Direct measurement of electrical charges on droplets.** Faraday cup measurements (Fig. 1c, Methods, Supplementary Section 2, and Supplementary Fig. 3) demonstrate that the excess positive charges on the water droplets dispensed from hydrophobic capillaries were proportional to the solid-liquid interfacial area. For instance, for the conical polypropylene capillaries, the excess electrical charges carried by five 10 μL droplets were greater than those carried by 50 μL droplets due to its conical geometry (Supplementary Fig. 4A). However, for the cylindrical FDTS-coated glass capillaries, the charges carried by five 10 μL droplets were equal to the charges carried by 50 μL droplets (Supplementary Fig. 4B). In addition, we found that the electrification of water droplets in our experiments was negligibly influenced by the air-water interfacial area (Supplementary Section 3, Supplementary Table 1). Furthermore, if the same polypropylene tip was used to consecutively load and dispense 300 droplets of 50 μL volume from a large water reservoir into the Faraday cup connected to the electrometer, the net charge or charge density of those droplets did not change significantly over time (Fig. 4c).

Streaming currents have been reported at high flow-speeds, such as >100 m s⁻¹, through capillaries of varied surface chemistries[13,14,77]. To investigate the effects of dewetting speed

relevant to our experiments, we withdrew 200 μL of water from an electroneutral water reservoir using polytetrafluoroethylene (PTFE) tubes of inner diameters 0.5 mm and 1 mm and then dispensed it at rates ranging from 0.06–2.4 mL min⁻¹ (speeds: 0.1–20.4 cm s⁻¹) inside the electrometer (Supplementary Fig. 5A). We found that for either tube the flow speed did not influence the magnitude of the total charge of the dispensed water droplets. However, the sum of the electrical charges carried by the dispensed water droplets from the 0.5 mm-wide capillary was approximately two-times larger than that from the 1 mm-wide capillary (Supplementary Fig. 5B) in agreement with the two-time higher interfacial area in the 0.5 mm-diameter tube than in the 1 mm-diameter tube. Note: the surface charge density was identical in the two PTFE tubes (Supplementary Fig. 5C). These results demonstrate that water electrification depends on the liquid-solid interfacial area.

**Origins of electrification at water–hydrophobe interfaces.** To investigate the origins of water electrification we considered whether: (i) the excess charge originated at the interface of the water and the hydrophobic surface due to, for instance, the selective adsorption/desorption of $H_3O^+$ or $OH^-$; or (ii) if the hydrophobic capillaries selectively drew water with excess charge during the intake process. To test these hypotheses, we placed a water reservoir inside a Faraday cage and monitored changes in its electrical charge during water withdrawal/addition. Specifically, we placed 1 mL of water (reservoir) inside the Faraday cage and we extracted aliquots (15–50 μL) using capillaries of varying wettability, while logging the charge of the water reservoir (Fig. 4a). Simultaneously, we added these extracted aliquots to another electrometer and measured the response. We found that when we extracted water using hydrophobic capillaries made of polypropylene, FDTS-coated glass, and polytetrafluoroethylene tubes, the charge on the bulk water reservoir became negative (Fig. 4b, red bars). Interestingly, when we added these extracted water aliquots to another electrometer using the same hydrophobic capillaries, we recovered equal and opposite positive charges (Fig. 4b, blue bars). From these data we deduced that polypropylene, FDTS-coated glass, and polytetrafluoroethylene surfaces have surface charge densities of $\sigma = -0.7 \pm 0.1$ nC cm⁻² (or $43 \pm 7$ # μm⁻²), $-0.46 \pm 0.11$ nC cm⁻² (or $29 \pm 7$ # μm⁻²), and $-0.12 \pm 0.04$ nC cm⁻² (or $7 \pm 2$ # μm⁻²), respectively, where # refers to the number of electronic charges (Eq. 5). From the chemical composition of these materials, we also know that they do not have Brønsted acid groups. In contrast, when we performed the same experiment using hydrophilic glass capillaries, which have silicic acid groups, we observed an opposite trend in the electrification, albeit with significantly lower magnitude in comparison to hydrophobic capillaries (Fig. 4b and the explanation presented in the Discussion section).

**Dependence on ionic strength and dielectric constant of aqueous solutions.** Next, we investigated the effects of ionic strength, water pH, dissolved $CO_2$ concentration, and dielectric constant of different aqueous mixtures on the electrification process. In dilute electrolytes containing simple monovalent salts, the characteristic length-scale at which electrostatic interactions persist, is known as the Debye length ($\kappa^{-1}$), and it is given by the formula

$$\kappa^{-1} = \sqrt{\frac{\varepsilon_r \varepsilon_0 k_B T}{2 N_A e^2 I}} \tag{6}$$

where $I$ is the ionic strength of the electrolyte, $\varepsilon_0$ is the vacuum permittivity, $\varepsilon_r$ is the relative permittivity of the medium, $k_B$ is the Boltzmann constant, $T$ is temperature in Kelvin, $N_A$ is Avogadro's number, and $e$ is the electronic charge[75]. We modulated the

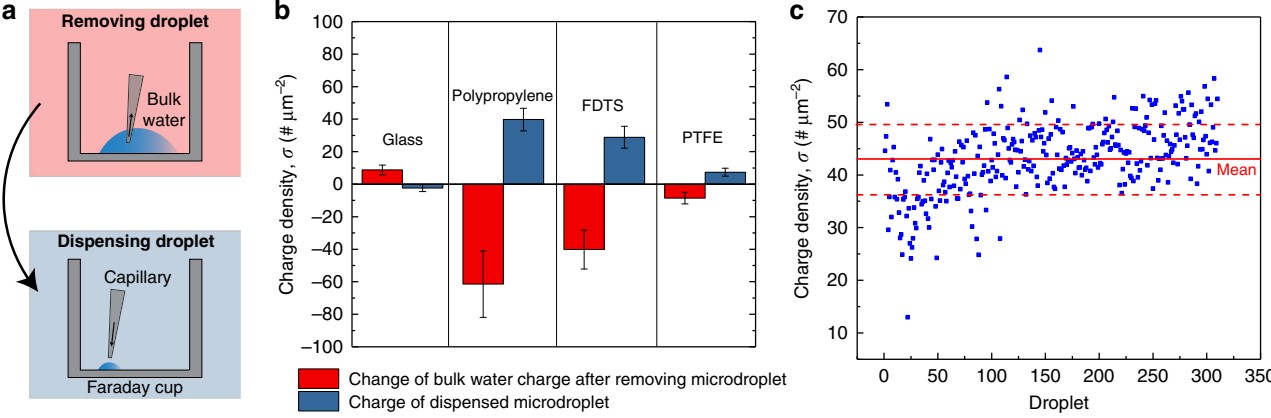

**Fig. 4 Quantification of the excess electrical charges carried by water droplets during pipetting. a** Removing droplet: we placed 1 mL of water (reservoir) inside a Faraday cup connected to an electrometer and extracted 15–50 μL aliquots at a time. Dispensing droplet: next, we dispensed the exctracted droplets into an electrometer to quantify their excess charges. We found that the charges incurred by the water reservoir and the withdrawn aliquots were equal and opposite. **b** Charge balance: charging of the water reservoir and the withdrawn aliquot using capillaries made of borosilicate glass, polypropylene, FDTS-coated glass, and PTFE. The charges on the water reservoir after withdrawing water aliquots using different capillaries are presented in red; commensurate (opposite) charges on the aliquots are presented in blue. (Note: the y-axis presents charge density, obtained by normalizing the charges by the liquid-solid interfacial area). Error bars represent the standard deviation of ten measurements. **c** Charge density of over 300 water droplets (50 μL each) loaded and dispensed from the same polypropylene capillary into an electrometer. The red dotted lines represent the ±1 standard deviations about the mean value of the surface charge density of polypropylene: $\sigma = -0.7 \pm 0.1$ nC cm$^{-2}$ or $43 \pm 7$ # μm$^{-2}$.

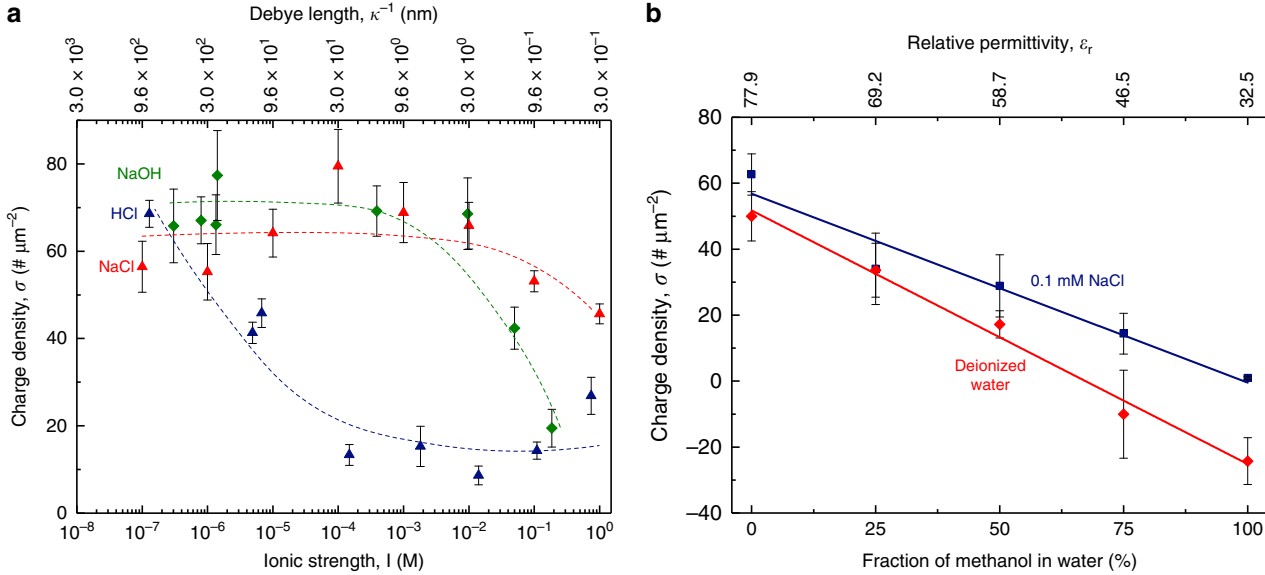

**Fig. 5 Electrification of water-hydrophobe interfaces as function of ionic strength, pH, and dielectric constant. a** Effects of varying Debye lengths on electrification at water-hydrophobe interfaces by varying ionic strengths with NaOH (green), HCl (blue), and NaCl (red). Aqueous electrolytes were drawn into polypropylene pipettes from charge-neutral reservoirs and dispensed into a Faraday cage connected to an electrometer to quantify electrical charges. The surface charge density was obtained by normalizing the observed charges by the solid-liquid interfacial areas inside the pipette prior to dispensing. (Dotted lines have been added to guide the eye.) **b** Effects of varying relative permittivity of liquids on the electrification at water-hydrophobe interfaces by varying dielectric constants by adding methanol to water and aqueous NaCl solutions (0.1 mM), using the same technique as in **a**. Methanol-rich mixtures demonstrate lower electrification in comparison to water-rich solutions (red datum points), and commensurate solutions with salty water (blue datum points). Error bars in each panel represent the standard deviation of ten measurements.

Debye lengths of aqueous solutions by varying the ionic strengths and relative permittivity.

We found that by suppressing the Debye lengths by adding ions, the charge carried by the pendant droplets dispensed from the hydrophobic capillaries decreased (Fig. 5a; see Supplementary Fig. 6A for the effects of varying ionic strength with KBr). Upon increasing the concentration of HCl, the measured charge dropped significantly below pH 4, whereas for NaCl and NaOH, the electrification was unaffected up until >10 mM (Fig. 5a).

Indeed, ion-specificity has been reported on various interfacial and bulk properties[26,78–80]. To probe whether the acidic solution had neutralized the negatively charged sites on the hydrophobic surface, we measured the change in the charge of the reservoir as well. We found that the charges accrued in the acidic reservoirs were significantly lower in magnitude (albeit, equal and opposite to those of the aliquots withdrawn) than those for NaCl, KBr, and NaOH cases (Supplementary Fig. 6B). This means that rather than the H$_3$O$^+$ ions getting permanently adsorbed to the water-

hydrophobe interface and neutralizing it, they were not even withdrawn from the reservoir in the first place. The lack of the permanent adsorption of the hydronium ions (or surface charge neutralization) is confirmed by the complete recovery of electrification when the same tips were used to pipette deionized water (Supplementary Fig. 6B). These results prove that lower electrification with acids does not always mean that they neutralize the surface charge as previously believed[11,38,40,48,81].

We lowered the dielectric constant of water ($\varepsilon_r = 78$) by adding controlled quantities of methanol ($\varepsilon_r = 32.5$). The electrification of the water-methanol mixtures decreased with the decreasing dielectric constants (Fig. 5b). Additionally, the electrification in methanol-rich solutions increased, if we increased the ionic strength by adding 0.1 mM NaCl (Fig. 5b, blue datum points).

Finally, we utilized diiodomethane ($CH_2I_2$) as a non-aqueous probe-liquid to investigate its electrification when brought into contact with polypropylene. We dissolved 1 mM NaCl in $CH_2I_2$; its surface tension, $\gamma \approx 51$ mN m$^{-1}$[82], facilitated the release of pendant droplets from the polypropylene capillaries at a flow rate ~3 mL min$^{-1}$. The dispensed droplets (with or without dissolved salt) did not have excess charge (Supplementary Fig. 7) because the low dielectric constant of diiodomethane, $\varepsilon_r = 5.3$[83], prevents charge separation during dispensing owing to the high electrostatic energy, $U_E \propto 1/\varepsilon_r$[75].

## Discussion

Here, based on our investigation of water-hydrophobe interfaces using the two complementary experimental techniques, we address the research questions raised in the introduction.

From our experimental results, we have deduced that the surfaces of solid hydrophobes such as polypropylene, or polytetrafluoroethylene (PTFE) or FDTS-coated glass are negatively charged even in air. When water comes into contact with these surfaces, the ions present in the water, such as $H_3O^+$, $OH^-$ and other cations/anions, form an electrical double layer with excess positive charges, and this charge separation is facilitated by the water's high dielectric constant. Thus, when a hydrophobic capillary draws water in from a reservoir, it draws in excess positive charges and leaves behind an equal and opposite negative excess charge at the source (Figs. 4, 6). Subsequently, when the water in this capillary is exposed as a pendant droplet to a uniform electric field, the excess positive charge causes the droplet to repel away from the positively charged plate (Figs. 2b, 3). On the other hand, if the capillary is water-filled case, the pendant droplet exchanges ions with the reservoir when the electric field is turned on, becomes slightly negatively-charged and deflects towards the positively charged plate (Fig. 2a and Supplementary Movie 1).

The absence (or presence) of linear scaling relationship between the excess charge density of the droplets, $\sigma$, and the applied electric field strength, $E$, for the air-filled (or water-filled) scenarios (Fig. 3) follows from equations [4–5]:

$$\sigma = q/A_o = mg \tan \alpha / EA_o \tag{7}$$

Variational analysis of Equation [7] yields (and derivation presented in Supplementary Section 4):

$$\Delta\alpha \propto \left[\frac{\Delta E}{E} + \frac{\Delta q}{q}\right] \sin(2\alpha) \tag{8}$$

Therefore, the change in the tilting angle, $\Delta\alpha$, depends on the change in the electric field strength, $\Delta E$, and the excess charge in the drop, $\Delta q$. For the air-filled capillary (Fig. 2b), $\Delta q = 0$ because no charge can flow into or out of the droplets. On the other hand, when the capillary is water-filled and/or connected to a reservoir (Fig. 2a, c), the higher electrical potential inside the capacitor

drives ionic current between the pendant droplet and the water reservoir and the ensuing charge $\Delta q \neq 0$ yields the observed scaling behavior between the charge density with the applied field (Fig. 3). Of course, this ion-exchange sets up a potential difference along the length of the capillary, which is why in the transitional case (Fig. 2c), the initial droplets attract towards the positive plate, but the latter ones repel away from it. Quantitative details of this process fall beyond the scope of this work. Additionally, this analysis does not account for the effects of the contact angle hysteresis at the capillary-water-air interface, the evaporation of water, and the non-uniform distribution of the excess charge in the droplets' bulk and at the air-water interface.

Dipole (size and moment) and ionic strength are the key properties of aqueous solutions that govern the electrification. Dipole characteristics determine their ability to dissociate ion pairs, while ionic strength determines the electrostatic screening length. The negative charge density on hydrophobic surfaces is the other crucial factor for electrification – in fact, Lowell & co-workers[70], Bard & co-workers[71], and Wang et al.[84], have suggested that common solid hydrophobes, such as PTFE and polypropylene, have surface-bound electrons. On the other hand, dielectrics with ionizable functional groups, such as silica[85], get deprotonated/protonated depending on the pH-p$K_a$, thereby creating an electrical double layer[86]. However, when water leaves the surface, those chemical groups are reprotonated/deprotonated to become electrically neutral. Thus, pendant drops pipetted from capillaries with Brønsted acid/base groups present non-significant charging (Fig. 4b).

We also explored the impact of environmental factors. First, we studied the effects of relative humidity (~0% and ~60%) using a glove-box with 99% $N_2$ atmosphere and found that electrification increased with increasing humidity, as has been noted before[11] (Supplementary Fig. 8). Second, we investigated the role of dissolved bicarbonate ions due to the dissolution of ambient $CO_2$ in water and the subsequent speciation[87]. We compared the electrification at water-polypropylene interfaces in two scenarios: water supersaturated with $CO_2$ gas (pH ≈ 4), and water in equilibrium with the atmosphere (pH ≈ 5.6), and found that the excess charges carried by the pendant water droplets decreased significantly as the water became more acidic due to the formation of carbonic acid similarly to the HCl solutions, as explained above (Fig. 5a, Supplementary Figs. 10 and 6B). Our findings do not disprove the role of bicarbonate ions in electrification, as multi-body effects in water may drive bicarbonates towards neutral/negatively-charged water-hydrophobe interfaces[88,89].

Lastly, Galembeck et al.[81], investigated the electrification at interfaces between insulators (e.g., PTFE/polyethylene) as a function of rubbing speed, pressure, and duration. Under these conditions, mechanoradicals form and they undergo electron transfers and self-assemble into microscale mosaics comprising hydrocarbocations and fluorocarbanions (and with surface charge densities >25-times observed in our work). These mobile charges can be removed by dissolving them into common polar and apolar solvents. To test whether the surface-bound charge in our experiments can be removed by solvents, we utilized polypropylene pipettes and measured the charges of the dispensed water droplets before and after washing them with acetone and methanol (Supplementary Fig. 9). We found that the average charge of the dispensed water decreased by ~20% in either case, yet, the final surface charge density of the pipettes remained stable even after four cycles of solvent washing and drying under our laboratory conditions. The 20% decrease in the charge density may be due to the partial solubility of the polymer surface. These results establish that the electrification that we observe is largely due to surface-bound immobile charges and sub-surface charges that are inaccessible to solvents. Systematic investigations are needed to unravel this.

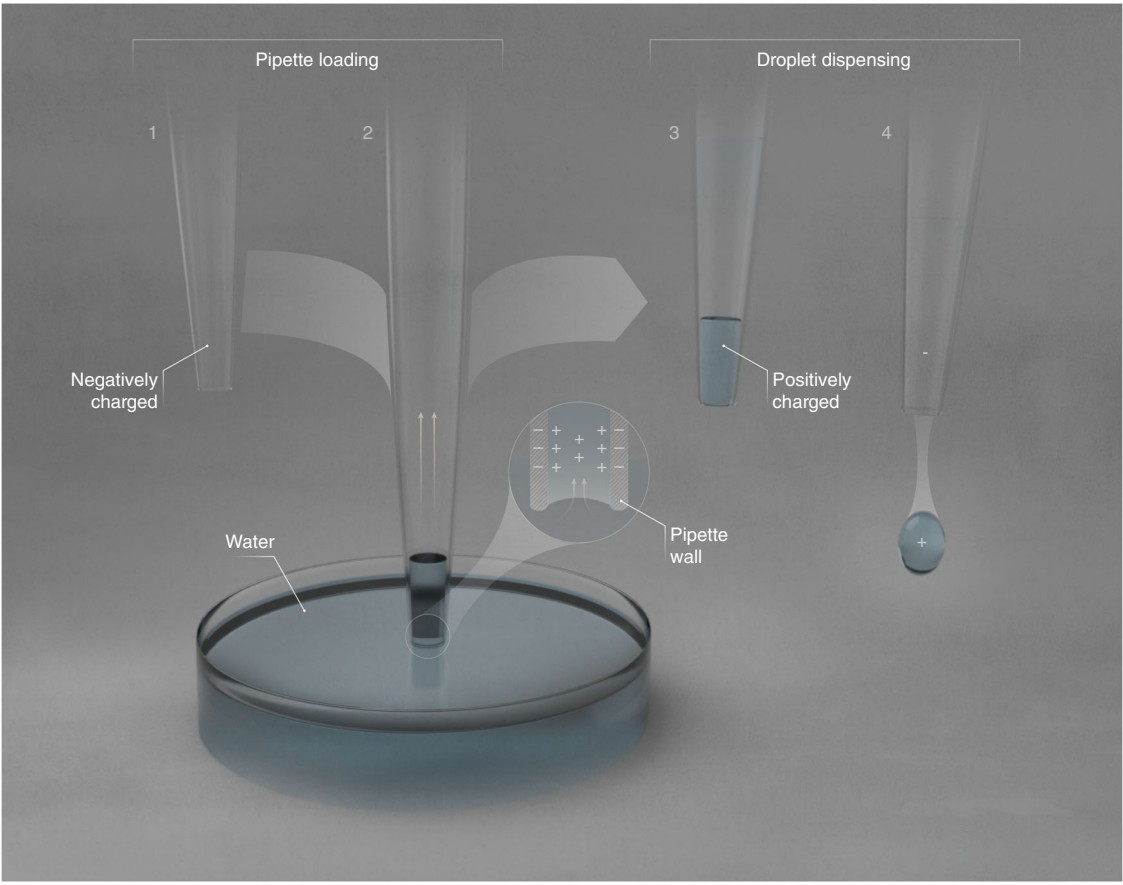

**Fig. 6 Proposed mechanism for the electrification at water-hydrophobe interfaces.** (1) Hydrophobic surfaces such as polypropylene, FDTS, and PTFE are intrinsically negatively charged. (2) When a hydrophobic capillary draws water in from an electrically neutral water reservoir, it selectively attracts cations that populate the electrical double layer (shown in the inset). (3) When the capillary is pulled out of the water reservoir, it carries water with a net-positive charge, for instance comprising $H_3O^+$ and other cations present, leaving behind an equal and opposite charge in the water reservoir. (4) When the water is dispensed from hydrophobic pipettes, hydrophobicity ensures that the entire volume exits the capillary, i.e., leaving no thin layer of water behind. Thus, the formed pendant droplet contains the excess positive charge that was drawn in originally, and it experiences repulsion from a positively charged plate as observed in our experiments (Fig. 2b).

Water is not the only liquid to be electrified when brought in contact with non-polar hydrophobic surfaces, or capillaries during pipetting, as evidenced by the electrification of water-alcohol mixtures with or without dissolved salts (Fig. 5b). Hydrophobicity ensures that when water droplets are dispensed from capillaries, they do not leave thin film behind, like in the case of hydrophilic surfaces[90,91]. Although nano- and microdroplets might remain on surface defects[92], presumably, containing a tiny fraction of the excess charge, most of the water exits the pipette along with the bulk of the excess charge. So, any liquid containing dissolved ions, high dielectric constant, and low wettability would experience electrification when brought into contact with a hydrophobic surface such as polypropylene, FDTS, and PTFE, among others.

While we have tried to disentangle various aspects of electrification at water/hydrophobe interfaces, our experiments cannot address all the remaining questions in this field. For instance, the origin of the negative charge of these surfaces, as deduced from our experiments, remains unclear at this point and should be explored; we refer the reader to the reports by Lowell & co-workers[70,93], Bard & co-workers[71,72], and Wang et al. [84]. We should specify that the conclusions drawn in this work about polypropylene, PTFE, and FDTS might not translate to all hydrophobic materials. Additionally, it is worth noting that AFM-based measurements of electrostatic repulsion between polypropylene surfaces in water at sub-30 nm separations reveal a

significantly higher surface charge density (~111 nC/cm²)[94] than ours. Further investigation is thus needed to systematically unravel the contributions of surface roughness, surface preparation and handling, and the ambient atmosphere in these experiments. The physical properties of liquids could also pose challenges in disentangling the factors underlying electrification. For instance, whereas mixing water with methanol reveals the dependence of interfacial charging on dielectric constants, it also lowers the surface tension, which makes the release of pendant droplets difficult; while diiodomethane's surface tension facilitates satisfactory detachment of pendant droplets, its low dielectric constant precludes charge separation. The unavoidable hydrolysis and cross-linking of silanes, manual dispensing, variations in the ambient conditions, left-over nano/micro droplets inside pipettes, and the polarization of capillaries inside the capacitor could contribute to experimental errors.

Our experimental results demonstrate that when water comes into contact with common hydrophobic materials such as polypropylene, FDTS, and PTFE, cations partition to the solid-liquid interface. Thus, these surfaces have a negative surface charge density. The electrical double layer theory predicts ensuing trends: liquids with lower dielectric constants and higher ionic strengths experience lower electrification. The role of hydrophobicity on droplet charging is limited to the fact that when water leaves the surface, it does so without leaving a liquid film

behind[92] containing electrical charges. These findings challenge the prevailing notions that the electrification of water-hydrophobe interfaces is driven exclusively by the dipolar arrangement of interfacial water, or the partial charge transfer between interfacial water and the hydrophobe, or due to the specific attachment/detachment of $OH^-$ or $H_3O^+$ ions at water-hydrophobe interfaces. We propose that any non-wetting liquid-solid interface, where the solid surface has a surface charge density and the liquid has dissolved ions and high dielectric constant, can cause droplet electrification. In principle, the electrification trends could reverse, e.g., from negative to positive, for a hydrophobic surface with positive charge. These findings advance the scientific understanding of electrification and rational design of technologies that involve electrowetting[1], micro/nano fluidics and pipetting[6,7,9,95], triboelectric power generation[10,12], desalination[96,97], and materials and surface engineering[91,98,99], among others.

## Methods

**Aqueous solutions.** We used MilliQ Advantage 10 (18.2 MΩ-cm, 3ppb) deionized water for this study. The water purification unit consisted of a Q-Gard pretreatment pack, UV lamp, Quantum cartridge (activated carbon and ion exchange resins) and a Q-Pod dispenser for final polishing[100]. We prepared electrolyte stock solutions of 1 M NaOH, HCl, and NaCl that we diluted to adjust ionic strengths. To saturate deionized water with $CO_2$, we bubbled the gas in it for 2 h, leading to pH ≈ 4 solutions.

**Capillaries.** Rigid hydrophobic substrates that we employed were: (1) polypropylene pipettes (Fisherbrand™ SureOne™ Micropoint, 02707430, 200 μL), (2) PTFE tubes (RCT Reichelt Chemietechnik GmbH + Co., 92543, inner diameter: 1 mm; 92529, inner diameter: 0.5 mm), and (3) cylindrical borosilicate glass capillaries (Sutter Instrument, BF150-110-10, outer diameter: 1.5 mm, 10 cm length) grafted with perfluorodecyltrichlorosilane (FDTS). The procedure of FDTS deposition was as following: we cleaned the capillaries with a fresh piranha solution ($H_2SO_4$: $H_2O_2$ = 4:1) for 10 min, followed by $O_2$ plasma activation (200 W, 16.5 sccm flow rate) for 10 min, and then the capillaries were placed in molecular vapor deposition system for coating using a 2-injection deposition cycle. (For details, please refer to ref. [101].) We determined the topography of the samples by Veeco Dimension Icon SPM (Supplementary Section 1, Supplementary Fig. 1). Samples were stored in sealed petri dishes in clean nitrogen-flow cabinet; please see ref. [102] for details.

**Advancing/receding contact angles in air.** We characterized the water-repellence of the capillaries by measuring advancing ($\theta_A$) and receding ($\theta_R$) contact angles of sessile deionized water droplets of volume ≈2 μL, dispensed/retracted at 0.2 μL s$^{-1}$ (Kruss Drop Shape Analyzer DSA100, Advance software). We found $\theta_A$ = 113° ± 2° and $\theta_R$ = 62° ± 1° on polypropylene, and $\theta_A$ = 105° ± 1° and $\theta_R$ = 72° ± 2° on FDTS-coated glass. The contact angles of water on oxygen-plasma treated glass capillaries were $\theta_A \approx \theta_R \approx 5°$.

**Permittivity of the solutions.** We measured the complex permittivity of the solutions using an open-ended coaxial probe method and radio-frequency vector network analyzer (300 kHz to 4.5 GHz range).

**Probing the electrification of pendant droplets under uniform electric fields.** We used a parallel plate capacitor (100 × 100 mm$^2$ aluminum plates) connected to a Keithley's 2290-10 high voltage source (range 0–10 kV). A Braintree Scientific BS-8000 syringe pump was employed to fill the capillaries with desired solutions and to form droplets at their tip. We recorded the deflections of the water droplets using a Phantom v1212 high-speed camera from Vision Research (Fig. 2a, b, Supplementary Movies 1 and 2). We used a Sony A5000 camera to record the rest of the experiments (Figs. 2c and 3b).

**Direct measurement of charge using an electrometer.** For the direct measurement of the charge, we employed a Keithley 6517B Electrometer (low input bias current <3 fA, high input impedance of 200 TΩ) connected to a homemade aluminum Faraday cup, which prevents the influence of external electrical sources on electrification. As a charged object reaches the cup, a charge of opposite polarity is induced on the inner electrode while the outer electrode is grounded, and the cup therefore acts as a capacitor. Due to a special low-current amplifier, the electrometer can detect very low charges by integrating the input current because the integrating capacitor is a part of the feedback loop (Fig. 1c)[76]. We recorded the charges using a LabVIEW program. We performed this part of the study at ~0% relative humidity in a glovebox (Cleatech, 2200-2-B).

## Data availability

The authors declare that all the data supporting the findings of this study are available within the paper and its Supplementary Information. Source data for the Figs. 3, 4b, 5, S4–S10 are provided. Source data are provided with this paper.

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

## Acknowledgements

H.M. and J.C.S. acknowledge funding from KAUST. J.N. and H.M. thank Mrs. Emilie Dauzon (KAUST), who initiated this research through investigating electrification at solid-solid interfaces in HM's Group as a visiting student in the year 2015. The co-authors thank Mr. Farizal Hakiki for performing permittivity measurements used in Fig. 5b, and Dr. Michael Cusack (KAUST) for scientific editing. The authors thank Mr. Ivan Gromicho, Scientific Illustrator at KAUST, for preparing Fig. 6.

## Author contributions

H.M. conceived the project, co-supervised the research with JCS, and coordinated the collaboration. H.M. and JCS conceived experiments detailed in Fig. 2A; J.N. and Z.S. planned the experiments reported in Fig. 2B; H.M. conceived the experiment reported in Fig. 2C; Z.S. and J.N. conceived the experiment detailed in Fig. 4; H.M. and A.G.J. conceived the experiment detailed in Fig. 5B. Z.S. and J.N. performed the experiments described in Figs. 2–3 and analyzed the data; J.N. performed the experiments reported in Figs. 4–5 and analyzed the data. M.I. and J.N. characterized purity of hexadecane using porous alumina membranes and interfacial tension measurements. H.M. derived the scaling relationships to explain Fig. 3; H.M., J.N., Z.S., J.C.S. and A.G.J. analyzed the data; H.M. and A.G.J. pieced together the results to realize the complete picture. H.M. wrote the paper and the co-authors edited it.

## Competing interests

The authors declare no competing interests.
