## [Peer Review File · Nature Communications]

Reviewers' comments:

Reviewer #1 (Remarks to the Author):

Review of Manuscript "Electrification at Water-Hydrophobe Interfaces" by Mishra and co-workers

What are the major claims of the paper?

This paper claims to show how the electrification of water droplets depends on hydrophobicity/hydrophilicity of the pipetting capillary surface (and amount of interface area).

Additionally, the authors claim that water is not the only liquid undergoing electrification and that electrification is not due to the usually discussed specific adsorption of OH⁻ and H₃O⁺.

Are they novel and will they be of interest to others in the community and the wider field?

The topic is of wider and practical interest in many different fields. The experiments used in this work are simple and appealing.

Is the work convincing, and if not, what further evidence would be required to strengthen the conclusions?

The evaluation and discussion is often vague and difficult to follow; statements remain rather speculative and are sometimes not logic:

1) Hydrophobic/hydrophilic is recognized as the key element, the possible mechanisms of surface charging during dewetting in presence of a double layer remain totally obscure

and no systematic dewetting experiments are described to address these really centrally interesting issues.

2) Strictly your experiments demonstrate excess ions in dispensed droplets and not the charging of surfaces left behind. You logically mix this up throughout.

3) The list of possible application areas in your introduction is not readily clear.

4) The use of trichlorosilanes to hydrophobize surfaces of glass yields highly undefined surfaces due to continued hydrolysis and crosslinking in the structure (ill-defined hydrophobic surface)

- 5) Your droplet shape analysis assumes a homogeneous surface charge, this assumption is oversimplified, how does it affect your results(?)
- 6) The capillaries in your experiment are exposed to the electrical field, which may be very relevant.
- 7) does the electrification effect depend on dewetting speed? - no mention about this in your paper
- 8) Figure 3B contains some of your most important experimental results; it is displayed too small and not discussed in detail.
It nicely illustrates how external electric fields (energy) can displace EDL ions beyond the Debye length.
- 9) What is the surface of the Faraday cup made of? This interface is equally important as the one of the capillary.
- 10) The mixture ratio of water/methanol will (linearly) affect solubility of EDL ions, just as shown in Fig 5B. Maybe the dielectric constant (your hypothesis) is irrelevant.
- 11) The mixture of water/methanol is not a "liquid other than water"; you make an unjustified statement about this in the conclusions.
- 12) You repeatedly and almost magically use the term "intrinsically negatively charged surfaces" and never define what you really mean. You just say that it is not (Bronsted) acid.
- 13) The section on Published Studies Revisited is a bit lengthy and not easy to read as it seems to be a diffuse gross attack on existing concepts without valid storyline nor counter-hypothesis.
- 14) Conclusions: No, your experiments do not show intrinsic negative charge - they show released droplet electrification (nice experiments), unfortunately without offering a plausible new hypothesis.
- 15) Your claim that there is no liquid film left behind (dewetting of hydrophobe) is pure speculation and lacks experimental evidence.

I think, the details of precisely this dewetting occurs in presence of a double layer and related charge separation/left behind is probably capable of explaining all your results.

On a more subjective note, do you feel that the paper will influence thinking in the field? Please feel free to raise any further questions and concerns about the paper.

I think the experimental approach is quite original and clearly shows that water droplet electrification is ion-based and depends on wetting-dewetting conditions.

I hope my comments can help to improve this seemingly preliminary work.

Manfred Heuberger

Reviewer #2 (Remarks to the Author):

This article deals with a phenomenon that has been known for more than a century but remains poorly understood. This attests to the challenge in unravelling the details of the electrification of water in contact with hydrophobic surfaces. One of the complications has been the difficulty in obtaining reliable reproducible experiments that can be used to evaluate various explanations of the effect. I find the conceptually simple experiments described here to be a very important contribution to the field as they clearly demonstrate that

- 1) The hydrophobic surfaces have a negative charge (even in air)
- 2) This results in water acquiring a positive charge when it is imbibed into the capillary
- 3) This is then manifest in positively charged droplets when the water is released from the capillaries
- 4) This effect is observed when the capillary is not filled with liquid

I think the clever experimental design and the quality of the results make these arguments very easy to prosecute and as such I expect that this work will be highly impactful in a number of fields from colloid science to industrial processing.

The realization and demonstration that the droplets exhibit fundamentally different behavior when they are produced by either an air filled dispenser or water filled dispenser is important. Some may say this is 'obvious', but I would say this is only the case in hindsight now that it has been so clearly demonstrated and explained, to my knowledge for the first time.

However, I do think the manuscript can be improved substantially and hope the authors choose to do so before the work is published.

Specifically

On Line 67 the authors state “Specifically, what are the roles of liquid dipole moment, pH, ionic strengths, dielectric constants, solid surface energy, water contact angles, surface-bound charges, relative humidity, ambient CO₂ concentration and ambient contamination?”

I think these claims cannot be substantiated and this text should be removed. Whilst the authors have performed some experiments where such parameters are varied the experiments have not been performed in a manner where the effect of these parameters can be separated from other effects. For example the effect of ionic strength remains unclear as the observed dependence of the specific nature of the ions is large so that the ionic strength cannot be used to describe the influence of electrolyte on charge density (see Figure 5A). Further, for NaCl the effect is not proportional to ionic strength. Similarly the addition of methanol has an effect and this is attributed to the change in dielectric properties of the solvent mixture, but this conclusion could only be verified by examining the effect of a different solvent mixture and showing that the charge density observed can be described by the dielectric constant of both mixtures.

The authors cannot claim to have systematically studied (with the exception of pH) any of the listed parameters and cannot provide a quantitative description of how they influence the charge on a droplet.

Later the authors state (line 360, pg 18), “The Debye length is the most important property of aqueous solutions in the context of electrification at water-hydrophobe interfaces.” Given this statement, it suggests that the authors should also study an additional electrolyte (eg KBr) that does not influence the pH to demonstrate this is indeed the case. It is a reasonable explanation but it would be good to establish this by additional experiments.

The data supplied is more relevant to the effect of pH. As such more discussion of the effect of HCl and NaOH would be helpful. The data suggests that the polymer surface is nearly neutral at pH values below 4 and increases with pH, being fully charged beyond pH 7 – and at very high NaOH concentrations the Debye length becomes important (when the surface charge is maintained). This all fits with the influence of pH and ionic strength on the double layer. Similarly, I think the effect of CO₂ observed is best described by the effect it has on pH rather than the effect on the ionic strength or the species produced. The discussion regarding bicarbonate (lines 383-387) needs to be revisited. The presence of the bicarbonate ions resulting from CO₂ dissolution is irrelevant as they will be repelled by the negatively charged surface, it is the H⁺ liberated (as shown by the drop in pH) when CO₂ dissolves, which will negate the negative surface charge at low pH. This is consistent with the data provided for HCl and indicates that the effect of CO₂ can be understood as arising from its acidic properties.

The clear suggestion from the data is that when a polymer surface is immersed in neat methanol the surface charge is removed (we still expect a double layer in methanol, albeit of reduced thickness). I believe that this is already known in the literature – (though I cannot find the specific example at the moment – perhaps in the work of Galembeck). This suggests an important additional experiment that could easily be performed without any additional materials or change to the experimental set-up. By first removing the charge on the polymer (or hydrophobic) surface using methanol the behavior of subsequent drops of water could be measured to determine how the (positive) charges on the droplets change with time. This would reveal the rate at which the negative charge on the polymer surface is regenerated whilst also confirming the importance of this negative charge. Is the recovery very fast or slow? Does it depend on humidity? Is it much quicker when immersed in water? I think such information will be important in understanding the charging mechanism of hydrophobic surfaces which remains a subject of great interest. Hence, I would really like to see this additional experiment added to the paper.

Minor comments:

Equation 3, why not replace L/V with $1/E$? Surely the electric field is the important parameter and the same electric field could be achieved with different value of L provided the L/V ratio was held constant.

Line 220 Should “diminish” be replaced with “change”? I suggest this as using the word diminish does not rule out the charge density increasing.

The explanation on page 18 for why the droplets expressed from a liquid reservoir are charged proportionally to the electric field is not completely clear to me. In this case there is a net positive charge on the liquid and the electric field is described as promoting ion exchange with the reservoir - presumably by repelling the positive charges (though isn't this direction perpendicular to the electric field). Could the authors describe precisely how this leads to drops that are negatively charged? One might think if the positive charge promotes exchange with the reservoir this process would cease when then the positive charges have been neutralized.

Sincerely,

Vincent S. J. Craig

Reviewer #3 (Remarks to the Author):

This manuscript presents experimental results aimed at understanding how different solid surfaces, primarily hydrophobic, influence the charge of interfacial water. The results are clearly described and thought provoking. I expect that these results will be quite interesting to the communities that study aqueous liquid-solid interfaces. However, I think the manuscript falls short of providing satisfying explanations for these results. Part of this is due to the rather ambitious claims made at the end of the introduction about the scope of the questions that the manuscript claims to answer (easily the topics of multiple separate studies), and part is due to the lack of consideration given to alternate possible explanations for the observed behavior. I think these results should be published but they belong under a more specialized journal title and without the use of overreaching overly general conclusions.

The principle conclusion is that hydrophobic surfaces carry an excess negative surface charge and that this charge promotes the accumulation of positive charge at the water interface. I find this explanation to be plausible, however, the extent to which this finding is general to hydrophobic surfaces or pertains to just the surface under investigation here is not adequately resolved.

The authors assume that hydrophobicity of the capillary ensures that no residual water film remains after the drop is ejected. Further testing should be done to confirm this point. Coulomb interactions are strong and I expect that if the interior of the capillary does indeed feature surface bound electrons, then those regions would be effectively hydrophilic and potentially strong enough to promote the sticking of the excess H_3O^+ .

Much of the interest and debate over this topic in the field has focused on unraveling the relative thermodynamic preference of H_3O^+ or OH^- to populate hydrophobic or air-water interfaces. By positing hydrophobic surface charge effects, insight into the native preference of OH^- or H_3O^+ for interfacial solvation is not advanced in this manuscript.

Furthermore, I find that the accumulation of negative charge in droplets in contact with hydrophilic surfaces, or for hydrophobic surfaces including a capillary water reservoir are not adequately explained.

I'm not sure data set (i) in Fig. 3B is convincingly constant across increasing field strengths. Since constant behavior is central to the conclusions of the manuscript, there would be benefit to working towards reducing the error bars in this data.

In Fig. 4C there appears to be systematic shift in the charge of dispensed drops consecutively over the first 100 drops. Why would this be?

We thank the reviewer (Prof. Manfred Heuberger) for his thorough and constructive assessment
of our manuscript. We have responded to each comment (in red) below in **bold text**, and we have
highlighted all of the corresponding changes that we have made in the revised manuscript. We
consider our revised manuscript to be appropriate for publication in *Nature Communications*.

**Reviewer #1.**

Comments:

This paper claims to show how the electrification of water droplets depends on
hydrophobicity/hydrophilicity of the pipetting capillary surface (and amount of interface area).
Additionally, the authors claim that water is not the only liquid undergoing electrification and that
electrification is not due to the usually discussed specific adsorption of OH⁻ and H₃O⁺.

The topic is of wider and practical interest in many different fields. The experiments used in this
work are simple and appealing.

The evaluation and discussion is often vague and difficult to follow; statements remain rather
speculative and are sometimes not logic.

**We thank the Reviewer for recognizing the importance of this work and raising excellent**
**points to help improve the quality of the manuscript. In response, we have substantially**
**revised the introduction, results, and discussion sections of the manuscript to enhance its**
**clarity; new experimental results have been added (Figs. S5, S6B, S7, S9 and additional**
**datum points to Fig. 3). We also attach an Unpublishable SI file containing new experimental**
**results that are relevant in answering Q1.1 and Q2.4.**

**Our point-by-point responses to the questions/comments follow.**

Q1.1 Hydrophobic/hydrophilic is recognized as the key element, the possible mechanisms of
surface charging during dewetting in presence of a double layer remain totally obscure
and no systematic dewetting experiments are described to address these really centrally interesting
issues.

**A1.1 Thank you for raising these points. The goal of this work is to demonstrate that when**
**water comes into contact with common hydrophobic surfaces such as polypropylene (PP), or**

polytetrafluoroethylene (PTFE) or FDTS-coated glass, cations present in water (e.g., H_3O^+)
form an electrical double layer at the water-hydrophobe interface. We have proved this by
two independent techniques:

- 1. Using a parallel plate capacitor, we have demonstrated that pendant droplets formed at
the tip of PP, or PTFE, or FDTS-coated glass capillaries repel from the positively charged
plate (Figures 1B, 2B). By investigating these deflections, we have estimated the excess
positive charge inside the droplets and the surface charge density of these materials that
drives the electrification (Please, refer to the air-filled cases in Figure 3).
- 2. Using an electrometer, we have demonstrated the presence of an excess positive charge
carried by water droplets withdrawn from electroneutral reservoirs using a PP, or PTFE,
or FDTS-coated glass capillary (Figure 4B, blue data points). The generation of an equal
and opposite negative charge in the reservoir due to this pipetting is also demonstrated
(Figure 4B, red data points).

The quantitative results from these two techniques are in excellent agreement. Based on
them, we have eliminated some of the highly debated hypotheses for the electrification of
water-hydrophobe interfaces, including water dipole moment,^{1, 2, 3} partial intramolecular
charge transfer within interfacial water^{4, 5} or between water and hydrophobes⁶, and the
specific adsorption of OH^- or H_3O^+ ions formed due to faster auto-dissociation of water⁷.

To further explore the electrification from the lens of the electrical double layer
theory, we probed the effects of (i) ionic strengths of aqueous solutions adjusted by HCl,
NaOH, and NaCl (Figure 5A) and (ii) dielectric constants of water-alcohol mixtures with 0.1
mM NaCl in the volumetric ratio 0-100% (Figure 5B). We found that the electrification
decreased with the increasing ionic strength and decreasing dielectric constant.

While we do not know the origins of the negative charge of these electrically insulating
and hydrophobic surfaces, the hypothesis of surface-bound electrons or “crypto electrons”
proposed by Bard & co-workers^{8, 9} seems pertinent; the latest report by Wang et al.,¹⁰ on the
electrification of water-insulator interfaces due to surface-bound electrons is quite relevant.
Hopefully, our report will advance this multidisciplinary research area.

Following the Reviewer’s comment, we performed two additional experiments to
investigate the effects of (i) liquid-solid surface area and flow speed while dispensing, and (ii)
liquid electric permittivity on the electrification. We describe them below:

(i) We withdrew 200 μL of water from an electroneutral water reservoir using
polytetrafluoroethylene (PTFE) tubes of inner diameters 0.5 mm and 1 mm and then
dispensed it at rates ranging from 0.06-2.4 mL/min (speeds: 0.1-20.4 cm/s) inside an
electrometer (Figure S5A). We found that for either tube this variation in the flow speed
did not influence the magnitude of the total charge of the dispensed water droplets.
Furthermore, the sum of the electrical charges carried by the dispensed water drops from
the 0.5 mm-wide capillary was approximately two-times larger than that from the 1 mm-
wide capillary (Figure S5B). It is due to the fact that for the same volume, $V = \pi r^2 l$, of
water, the liquid-solid interfacial area, $A = 2\pi r l$, in the 0.5 mm-diameter tube was two-
71 times larger than that in the 1 mm-diameter tube. This simple experiment demonstrates
that the electrification depends on the liquid-solid interfacial area. The surface charge
density in either case was identical (Figure S5C), as both tubes were composed of PTFE.

(ii) In our experiments to investigate the effects of the dielectric constant on the
electrification of water-methanol mixtures, we had noted that as the relative permittivity
decreased from 78 to 32.5 from pure water to pure methanol, the electrification decreased
(Figure 5B). Towards a non-aqueous solution, we utilized diiodomethane with surface
tension, $\gamma_{LV} = 50.8 \text{ mN/m}$, and dielectric constant, $\epsilon_r \approx 5$ at normal temperature and
pressure, which could dissolve 1mM NaCl. We found that when droplets of
diiodomethane (with or without 1mM NaCl) are manually dispensed from polypropylene
capillaries at the rate of $\sim 3 \text{ mL/min}$, they do not demonstrate measurable electrification.
We consider this to be due to the low dielectric constant, which prevents charge
separation during the dispensing of drops owing to the high electrostatic energy for
charge separation, $U_E \propto 1/\epsilon_r$.

Changes made to the manuscript:

Lines 209-221: “Streaming currents have been reported at high flow-speeds, such as $> 100 \text{ ms}^{-1}$,
through capillaries of varied surface chemistries.^{11, 12, 13} To investigate the effects of dewetting
speed relevant to our experiments, we withdrew 200 μL of water from an electroneutral water
reservoir using polytetrafluoroethylene (PTFE) tubes of inner diameters 0.5 mm and 1 mm and
then dispensed it at rates ranging from 0.06-2.4 mL/min (speeds: 0.1-20.4 cm/s) inside the
electrometer (Figure S5A). We found that for either tube the flow speed did not influence the

magnitude of the total charge of the dispensed water droplets. However, the sum of the electrical
charges carried by the dispensed water droplets from the 0.5 mm-wide capillary was approximately
two-times larger than that from the 1 mm-wide capillary (Figure S5B) in agreement with the two-
time higher interfacial area in the 0.5 mm-diameter tube than in the 1 mm-diameter tube. Note: the
surface charge density was identical in the two PTFE tubes (Figure S5C). These results
demonstrate that water electrification depends on the liquid-solid interfacial area.”

**Figure S5.** (A) Schematics of the experimental set-up: the known volume of water was withdrawn
by the tube and subsequently pushed out by a column of air. The charge of the water was measured
by electrometer connected to the Faraday cup. (B) Correlation between the total electrical charge
carried by 200 μL water dispensed from PTFE tubes of inner diameters of 0.5 mm (red dots) and
1 mm (blue dots) as a function of the rate of dispensing (controlled by a syringe pump). The larger
liquid-solid contact area in the tube of 0.5 mm diameter led to higher electrical charging in
comparison with the tube with 1 mm diameter. (C) When the total charge in either scenario is
normalized by the liquid-solid interfacial area, similar charge density is obtained. This is expected
because the material composition of the tubes is the same.

**Lines 310-316:** “Finally, we utilized diiodomethane (CH₂I₂) as a non-aqueous probe-liquid to
investigate its electrification when brought into contact with polypropylene. We dissolved 1 mM
NaCl in CH₂I₂; its surface tension, $\gamma \approx 51$ mN/m,¹⁴ facilitated the release of pendant droplets from
the polypropylene capillaries at a flow rate ~ 3 mL/min. The dispensed droplets (with or without
dissolved salt) did not have excess charge (Figure S7) because the low dielectric constant of

diiodomethane, $\epsilon_r = 5.3$,¹⁵ prevents charge separation during dispensing owing to the high
electrostatic energy, $U_E \propto 1/\epsilon_r$.¹⁶”

**Figure S7.** Electrical charges of single drops of diiodomethane, diiodomethane containing 1mM
NaCl, water, and water containing 1mM NaCl, manually dispensed from polypropylene capillaries
at the rate of ~ 3 mL/min. Due to the low dielectric constant of diiodomethane ($\epsilon_r = 5.3$), even the
addition of 1 mM salt did not enhance its electrification during pipetting. In contrast, significantly
higher electrification was observed at the water-polypropylene interface due to water’s intrinsic
ions and higher dielectric constant ($\epsilon_r = 80$). Addition of 1mM NaCl increased the electrification
for water.

We also attach an **Unpublishable SI file (Section USI-1)** reporting some new
experimental results on the release of ions from the electrical double layer in the absence of
external electrical field for the Reviewers and the Editor. Briefly, the experiments entail
drawing water into a long PTFE tube from an electroneutral water reservoir and dispensing
it gradually as drops and recording their charges using an electrometer. The electrical
charges of all the drops are positive and the magnitude of the dispensed drops increases from
the beginning to the end. We think that this preliminary experiment demonstrates the

“release” of ions from the EDL, giving rise to a vertically-aligned electric field that pushes
the EDL ions downwards. However, a comprehensive study of this phenomenon is beyond
the scope of this report.

Q1.2 Strictly your experiments demonstrate excess ions in dispensed droplets and not the
charging of surfaces left behind. You logically mix this up throughout.

A1.2 Thank you for pointing this out. In the revised version, we clearly state what we
observed in the experiments and what we deduced from it. Changes made to the manuscript:

**Lines 26-33:** “Based on these results, we deduced that common hydrophobic materials possess
surface-bound negative charge, for example, polypropylene has a surface charge density of $\sigma = -$
0.7 ± 0.1 nC-cm⁻², i.e. the number of electronic charges (#) per μm^2 of 43 ± 7 #/ μm^2 . Thus, when
these surfaces are submerged in water, hydrated cations (e.g., H₃O⁺) form an electrical double
layer. Furthermore, we demonstrate that the primary role of hydrophobicity is to facilitate water-
substrate separation without leaving a significant amount of liquid (as films or droplets) behind.”

**Lines 238-242:** “From these data we deduced that polypropylene, FDTS-coated glass, and
polytetrafluoroethylene surfaces have surface charge densities of $\sigma = -0.7 \pm 0.1$ nC-cm⁻² (or 43
± 7 #/ μm^2), -0.46 ± 0.11 nC-cm⁻² (or 29 ± 7 #/ μm^2), and -0.12 ± 0.04 nC-cm⁻² (or 7 ± 2
#/ μm^2), respectively. From the chemical composition of these materials, we also know that they
do not have Brønsted acid groups.”

**Lines 325-327:** “From our experimental results, we have deduced that the surfaces of solid
hydrophobes such as polypropylene, or polytetrafluoroethylene (PTFE) or FDTS-coated glass are
negatively charged even in air.”

**Lines 451-461:** “Our experimental results demonstrate that when water comes into contact with
common hydrophobic materials such as polypropylene, FDTS, and PTFE, cations partition to the
solid-liquid interface. Thus, these surfaces have a negative surface charge density. The electrical
double layer theory predicts ensuring trends: liquids with lower dielectric constants and higher
ionic strengths experience lower electrification. The role of hydrophobicity on droplet charging is
limited to the fact that when water leaves the surface, it does so without leaving a liquid film

behind¹⁷ containing electrical charges. These findings challenge the prevailing notions that the
electrification of water-hydrophobe interfaces is driven exclusively by the dipolar arrangement of
interfacial water, or the partial charge transfer between interfacial water and the hydrophobe, or
due to the specific attachment/detachment of OH⁻ or H₃O⁺ ions at water-hydrophobe interfaces.”

**Q1.3 The list of possible application areas in your introduction is not readily clear.**

**A1.3 Thank you for this point. We have drilled down on the specific applications/examples**
**in the revised manuscript in the introduction as well as in conclusion.**

**Changes made:**

**Lines 42-47:** “The electrification of water against hydrophobic surfaces (hereafter referred to as
“water-hydrophobe” interfaces) plays an important role in various applied and natural contexts,
such as pipetting^{18, 19, 20}, triboelectric power generation^{21, 22, 23}, hydrogen generation^{11, 13},
mitigating dust deposition on solar panels,²⁴ preventing fire hazards in granular flows,²⁵ and
precipitation and thundercloud charging^{26, 27}.”

**Lines 465-468:** “These findings advance the scientific-understanding of electrification and rational
design of technologies that involve desalination,²⁸ electrowetting,²⁹ surface engineering,^{30, 31}
triboelectric power generation,^{21, 23} among others.”

**Q1.4 The use of trichlorosilanes to hydrophobize surfaces of glass yields highly undefined surfaces**
**due to continued hydrolysis and crosslinking in the structure (ill-defined hydrophobic surface)**

**A1.4 Thank you for raising this point. In the revised manuscript, we list this drawback of**
**silanes as a potential source of experimental error, among others.**

**Lines 445-448:** “The unavoidable hydrolysis and cross-linking of silanes, manual dispensing,
variations in the ambient conditions, left-over nano/micro droplets inside pipettes, and the
polarization of capillaries inside the capacitor could contribute to experimental errors.”

Q1.5 Your droplet shape analysis assumes a homogeneous surface charge, this assumption is
oversimplified, how does it affect your results(?)

A1.5 Thank you for raising this important question. Below, we explain why excess charge
carried by droplets is not “surface charge”:

(i) The charge densities reported in Figures 3, 4, 5, S5-S10 were obtained by normalizing q
by the liquid-solid area inside the capillary (not the air-water interfacial area of the
pendant drops). This is explained in lines 125-128, lines 184-186, lines 198-203, lines 216-
221, lines 254-256, and lines 302-304.

(ii) The horizontal force, F_E , that a pendant droplet experiences due to the uniform electric
field inside the parallel plate capacitor, E , is given by $F_E = qE$, where q is the net charge
inside the droplet. The deflection of the drops, which corresponds to the force balance
between F_E and the drop weight, allows us to quantify q . Crucially, the fractionation of
this charge, q , – inside the drop or at the air-water interface – is beyond the scope of
this work; it is a problem of much current interest in chemical physics^{32, 33}.

(iii) Lastly, in the Supplementary Section S3 and Table S1 we have demonstrated that the
electrification at the air-water interface, if at all, is negligible compared to that at the
interfaces of water with polypropylene, FDTS, and PTFE.

Q1.6 The capillaries in your experiment are exposed to the electrical field, which may be very
relevant.

A1.6 Thank you for this comment. The results presented in Figures 3, 4, and 5 are consistent
with each other even though they were obtained through two entirely different techniques:

(i) deflection of charged drops inside a parallel plate capacitor, and (ii) direct charge
measurement using an electrometer. Thus, the polarization effects might not significantly
alter the properties of (hydrophobic) insulators, e.g., polypropylene, FDTS, and PTFE.
However, in the revised version, we have listed this as a potential source for error:

Lines 445-448: “The unavoidable hydrolysis and cross-linking of silanes, manual dispensing,
variations in the ambient conditions, left-over nano/micro droplets inside pipettes, and the
polarization of capillaries inside the capacitor could contribute to experimental errors.”

Q1.7 Does the electrification effect depend on dewetting speed? - no mention about this in your
paper

A1.7 Thank you for this question. Following the Reviewer's question, we performed a series
of systematic experiments to investigate the effects of the flow speed on the electrification.
To this end, we gradually withdrew 200 μL of pure water from an electroneutral water
reservoir using polytetrafluoroethylene (PTFE) tubes of inner diameters 0.5 mm and 1 mm,
which was then dispensed at rates ranging from 0.06-2.4 mL/min (speeds: 0.1-20 cm/s) inside
an electrometer (Figure S5A). We found that for either tube this variation in the flow speed
did not influence the magnitude of the total charge of the dispensed water droplets. We
should note that we are aware of the work of Faubel¹¹ and Sakyally & co-workers¹² on
pressure-driven shear flows through capillaries at speeds in the range 50-400 ms^{-1} , leading
to the formation of H_2 gas. Physics underlying those systems, however, is an active area of
research.¹³

Changes made to the manuscript:

Lines 209-221: "Streaming currents have been reported at high flow-speeds, such as $> 100 \text{ ms}^{-1}$,
through capillaries of varied surface chemistries.^{11, 12, 13} To investigate the effects of dewetting
speed relevant to our experiments, we withdrew 200 μL of water from an electroneutral water
reservoir using polytetrafluoroethylene (PTFE) tubes of inner diameters 0.5 mm and 1 mm and
then dispensed it at rates ranging from 0.06-2.4 mL/min (speeds: 0.1-20.4 cm/s) inside the
electrometer (Figure S5A). We found that for either tube the flow speed did not influence the
magnitude of the total charge of the dispensed water droplets. However, the sum of the electrical
charges carried by the dispensed water droplets from the 0.5 mm-wide capillary was approximately
two-times larger than that from the 1 mm-wide capillary (Figure S5B) in agreement with the two-
time higher interfacial area in the 0.5 mm-diameter tube than in the 1 mm-diameter tube. Note: the
surface charge density was identical in the two PTFE tubes (Figure S5C). These results
demonstrate that water electrification depends on the liquid-solid interfacial area."

 **Figure S5.** (A) Schematics of experimental set-up: a known volume of water was withdrawn using
 a PTFE tube and subsequently pushed out by a column of air. Charges carried by water drops were
 measured by a Faraday cup connected to an electrometer. (B) Correlation between the total
 electrical charge carried by 200 μL water dispensed from PTFE tubes of inner diameters of 0.5
 260 mm (red dots) and 1 mm (blue dots) as a function of the rate of dispensing (controlled by a syringe
 pump). The larger liquid-solid contact area in the tube of 0.5 mm diameter led to higher electrical
 charging in comparison with the tube with 1 mm diameter. (C) When the total charge in either
 scenario is normalized by the liquid-solid interfacial area, similar charge density is obtained. This
 is expected because the material composition of the tubes is the same.

 Q1.8 Figure 3B contains some of your most important experimental results; it is displayed too
 small and not discussed in detail.

It nicely illustrates how external electric fields (energy) can displace EDL ions beyond the Debye
 length.

A1.8 Thank you. In the revised version, we have increased the size of Figure 3 for the readers'
 comfort. We also note that the results presented in this crucial figure have been discussed in
 lines 165-193 (the air-filled case) and lines 354-372 (the water-filled case).

 Q1.9 What is the surface of the Faraday cup made of? This interface is equally important as the
 one of the capillary.

A1.9 Thank you for raising this point. The Faraday cup was made of aluminum sheet.
 Changes to the manuscript:

**Lines 91-94:** “The second technique deployed to quantify the electrification of water used an
ultrasensitive electrometer (with a detection limit of 10 fC) equipped with a Faraday cup **made of**
**aluminum sheet** to shield external interferences³⁴ (Figure 1C, Figure S3, Methods).”

**Q1.10** The mixture ratio of water/methanol will (linearly) affect solubility of EDL ions, just as
shown in Fig 5B. Maybe the dielectric constant (your hypothesis) is irrelevant.

**A1.10** Thank you for this comment. At standard temperature and pressure conditions (STP:
293 K and 1 atm), the solubility limits of NaCl in methanol and water are ≈ 250 mM and
≈ 6000 mM, respectively.³⁵ In contrast, the maximum salt concentration in the experiments
pertaining to Figure 5B was 0.1 mM. Thus, we respectfully disagree with the reviewer’s point
that the solubility of EDL ions would vary significantly in water/methanol mixtures at such
a low concentration (well below solubility limit in pure methanol). In fact, in our response to
the following question (Q1.11), we discuss results obtained from new experiments that
bolster our hypothesis about dielectric constants.

**Q1.11** The mixture of water/methanol is not a "liquid other than water"; you make an unjustified
statement about this in the conclusions.

**A1.11** Thank you. In our investigation of the effects of relative permittivity on the
electrification of water-methanol mixtures, we had noted that as the relative permittivity
decreased from 78 to 32.5 from pure water to pure methanol, the electrification decreased
**(Figure 5B)**. Following the comments of Reviewer#1 and 2, we utilized diiodomethane as a
probe liquid: its high surface tension of $\gamma_{LV} = 51$ mN/m at STP facilitated satisfactory
detachment of pendant drops from the capillaries; its polarity and dielectric constant $\epsilon_r \approx$
5 at STP afforded the dissolution of 1mM NaCl in it. When 50 μ L droplets of diiodomethane
(with or without 1mM NaCl) were manually dispensed from polypropylene capillaries at the
rate of ~ 3 mL/min, they did not exhibit measurable electrification. We consider this to be
due to the low dielectric constant of the solvent, which prevents charge separation owing to
the high electrostatic penalty, $U_E \propto 1/\epsilon_r$.

**Lines 310-316:** “Finally, we utilized diiodomethane (CH_2I_2) as a non-aqueous probe-liquid to
investigate its electrification when brought into contact with polypropylene. We dissolved 1 mM
NaCl in CH_2I_2 ; its surface tension, $\gamma \approx 51$ mN/m,¹⁴ facilitated the release of pendant droplets from

the polypropylene capillaries at a flow rate ~ 3 mL/min. The dispensed droplets (with or without
dissolved salt) did not have excess charge (Figure S7) because the low dielectric constant of
diiodomethane, $\epsilon_r = 5.3$,¹⁵ prevents charge separation during dispensing owing to the high
electrostatic energy, $U_E \propto 1/\epsilon_r$.¹⁶”

**Lines 461-465:** “We propose that any non-wetting liquid-solid interface, where the solid surface
has a surface charge density and the liquid has dissolved ions and high dielectric constant, can
cause droplet electrification. In principle, the electrification trends could reverse, e.g., from
negative to positive, for a hydrophobic surface with positive charge.”

**Q1.12** You repeatedly and almost magically use the term "intrinsically negatively charged
surfaces" and never define what you really mean. You just say that it is not (Bronsted) acid.

**A1.12** From our experimental results (Figure 4), we have deduced that common hydrophobic
materials, which also happen to electrical insulators, are negatively charged (even in air).
We do know from the chemical make-up of these materials that they do not have Brønsted
acids as functional groups. Beyond that, we do not know about the origins of this negative
charge. However, we bring to the readers’ attention the reports by Lowell & co-workers,
Bard & co-workers^{8, 9} and Wang & co-workers,¹⁰ who have suggested surface-bound
electrons to be responsible for electrification. These points are now presented more lucidly
in the manuscript:

**Lines 238-242:** “From these data we deduced that polypropylene, FDTS-coated glass, and
polytetrafluoroethylene surfaces have surface charge densities of $\sigma = -0.7 \pm 0.1$ nC-cm⁻² (or 43
± 7 #/ μm^2), -0.46 ± 0.11 nC-cm⁻² (or 29 ± 7 #/ μm^2), and -0.12 ± 0.04 nC-cm⁻² (or 7 ± 2
#/ μm^2), respectively. From the chemical composition of these materials, we also know that they
do not have Brønsted acid groups”

**Lines 324-326:** “From our experimental results, we have deduced that the surfaces of solid
hydrophobes such as polypropylene, or polytetrafluoroethylene (PTFE) or FDTS-coated glass are
negatively charged even in air.”

**Lines 378-382:** “The negative charge density on hydrophobic surfaces is the other crucial factor
for electrification – in fact, Lowell & co-workers,³⁶ Bard & co-workers⁸, and Wang et al.,¹⁰ have
suggested that common solid hydrophobes, such as PTFE and polypropylene, have surface-bound
electrons.”

**Lines 435-439:** “For instance, the origin of the negative charge of these surfaces, as deduced from
our experiments, remains unclear at this point and should be explored; we refer the reader to the
reports by Lowell & co-workers^{36,37}, Bard & co-workers^{8,9}, and Wang et al.¹⁰. We should specify
that the conclusions drawn in this work about polypropylene, PTFE, and FDTS might not translate
to all hydrophobic materials”

**Q1.13** The section on Published Studies Revisited is a bit lengthy and not easy to read as it seems
to be a diffuse gross attack on existing concepts without valid storyline nor counter-hypothesis.

**A1.13** Thank you. Reviewer#2 also suggested that this section was not adding significant
value to the manuscript. After consulting the Editor, we have removed it.

**Q1.14** Conclusions: No, your experiments do not show intrinsic negative charge - they show
released droplet electrification (nice experiments), unfortunately without offering a plausible new
hypothesis.

**A 1.14** Thank you. We have addressed this point in A1.2.

**Q1.15** Your claim that there is no liquid film left behind (dewetting of hydrophobe) is pure
speculation and lacks experimental evidence. I think, the details of precisely this dewetting occurs
in presence of a double layer and related charge separation/left behind is probably capable of
explaining all your results.

**A1.15** Thank you for raising this excellent point. The Reviewer is correct in pointing out that
tiny micro/nano droplets might be left behind inside the hydrophobic capillary. In the revised
version, we list this as a potential source of experimental error. However, it should be realized
that as water is drawn inside a hydrophobic capillary from an electroneutral reservoir, it
brings in excess positive ions and the reservoir acquires a negative charge (Figure 4A-C).
The fact that those two charges are equal and opposite proves that the residual water on the

capillary surface is not a significant source of error in our experiments or their
interpretation.

**Changes to the manuscript:**

**Lines 423-427:** “Hydrophobicity ensures that when water droplets are dispensed from capillaries,
they do not leave thin film behind, like in the case of hydrophilic surfaces.^{30, 38} Although nano-
and microdroplets might remain on surface defects,¹⁷ presumably, containing a tiny fraction of
the excess charge, most of the water exits the pipette along with the bulk of the excess charge.”

**Lines 445-448:** “The unavoidable hydrolysis and cross-linking of silanes, manual dispensing,
variations in the ambient conditions, left-over nano/micro droplets inside pipettes, and the
polarization of capillaries inside the capacitor could contribute to experimental errors.”

**Q 1.16** I think the experimental approach is quite original and clearly shows that water droplet
electrification is ion-based and depends on wetting-dewetting conditions. I hope my comments can
help to improve this seemingly preliminary work.

**A 1.16** Thank you very much for your constructive criticism which has helped us improve
the manuscript substantively. We hope that you will find the revised manuscript package to
be worthy of publication in *Nature Communications*.

We thank the reviewer (Professor Vincent Craig) for his thorough and constructive assessment of
our manuscript. We have responded to each reviewer comment (in red) below in **bold text**, and
we have highlighted all of the corresponding changes that we have made to the attached
manuscript. We consider our revised manuscript appropriate for publication in *Nature*
*Communications*.

**Reviewer #2.**

**Comments:**

This article deals with a phenomenon that has been known for more than a century but remains
poorly understood. This attests to the challenge in unravelling the details of the electrification of
water in contact with hydrophobic surfaces. One of the complications has been the difficulty in
obtaining reliable reproducible experiments that can be used to evaluate various explanations of
the effect. I find the conceptually simple experiments described here to be a very important
contribution to the field as they clearly demonstrate that

1) The hydrophobic surfaces have a negative charge (even in air)

2) This results in water acquiring a positive charge when it is imbibed into the capillary

3) This is then manifest in positively charged droplets when the water is released from the
capillaries

4) This effect is observed when the capillary is not filled with liquid

I think the clever experimental design and the quality of the results make these arguments very
easy to prosecute and as such I expect that this work will be highly impactful in a number of
fields from colloid science to industrial processing.

The realization and demonstration that the droplets exhibit fundamentally different behavior
when they are produced by either an air filled dispenser or water filled dispenser is important.

Some may say this is 'obvious', but I would say this is only the case in hindsight now that it has
been so clearly demonstrated and explained, to my knowledge for the first time.

However, I do think the manuscript can be improved substantially and hope the authors choose
to do so before the work is published.

**We are grateful to the Reviewer for recognizing the importance of our contribution and**
**raising excellent questions in a collegial spirit. In response, we have substantially revised the**
**introduction, results, and discussion sections of the manuscript to enhance its clarity; new**

experimental results have been added (Figs. S5, S6B, S7, S9 and additional datum points to
Fig. 3). We also attach an Unpublishable SI file containing new experimental results that are
relevant in answering Q1.1 (Section USI-1) and Q2.4 (Section USI-2). Our point-by-point
responses to the questions/comments follow.

Q2.1 On Line 67 the authors state “Specifically, what are the roles of liquid dipole moment, pH,
ionic strengths, dielectric constants, solid surface energy, water contact angles, surface-bound
charges, relative humidity, ambient CO₂ concentration and ambient contamination?

I think these claims cannot be substantiated and this text should be removed. Whilst the authors
have performed some experiments where such parameters are varied the experiments have not
been performed in a manner where the effect of these parameters can be separated from other
effects. For example the effect of ionic strength remains unclear as the observed dependence of
the specific nature of the ions is large so that the ionic strength cannot be used to describe the
influence of electrolyte on charge density (see Figure 5A). Further, for NaCl the effect is not
proportional to ionic strength. Similarly the addition of methanol has an effect and this is attributed
to the change in dielectric properties of the solvent mixture, but this conclusion could only be
verified by examining the effect of a different solvent mixture and showing that the charge density
observed can be described by the dielectric constant of both mixtures.
The authors cannot claim to have systematically studied (with the exception of pH) any of the
listed parameters and cannot provide a quantitative description of how they influence the charge
on a droplet.

**A2.1 Thank you. We address your question in three parts:**

(i) We agree with the Reviewer. In response, lines 62-63 have been edited as per his
suggestion.

**Revised lines 62-63:** “How do the properties of aqueous solutions, solid surfaces and the
environment impact the electrification of water at water-hydrophobe interfaces?”

(ii) Indeed, the specific nature of ions and their ionic strengths on the interfacial
electrification – a problem of much current interest in chemical physics^{32, 33} – is beyond

the scope of this work. Following the Reviewer’s comments, we have added the following
to the discussion.

**Lines 273-291:** “We found that by suppressing the Debye lengths by adding ions, the charge
carried by the pendant droplets dispensed from the hydrophobic capillaries decreased (Figure 5A;
see Figure S6A for the effects of varying ionic strength with KBr). Upon increasing the
concentration of HCl, the measured charge dropped significantly below pH 4, whereas for NaCl
and NaOH, the electrification was unaffected up until > 10 mM (Figure 5A). Indeed, ion-
specificity has been reported on various interfacial and bulk properties^{32, 39, 40, 41}. To probe whether
the acidic solution had “neutralized” the negatively charged sites on the hydrophobic surface, we
measured the change in the charge of the reservoir as well. We found that the charges accrued in
the acidic reservoirs were significantly lower in magnitude (albeit, equal and opposite to those of
the aliquots withdrawn) than those for NaCl, KBr, and NaOH cases (Figure S6B). This means that
rather than the H₃O⁺ ions getting permanently adsorbed to the water-hydrophobe interface and
“neutralizing” it, they were not even withdrawn from the reservoir in the first place. The lack of
the permanent adsorption of the hydronium ions (or surface charge neutralization) is confirmed by
the complete recovery of electrification when the same tips were used to pipette deionized water
(Figure S6). These results prove that lower electrification with acids does not always mean that
they “neutralize” the surface charge as previously believed.^{7, 22, 42, 43, 44} We surmise that the higher
mobility of protons in water screens the electrical surface charge more effectively than hydrated
cations.”

**(iii)**In our experiments to investigate the effects of relative permittivity on the electrification
of water-methanol mixtures, we had noted that as the dielectric constant decreased from
78 to 32.5 from pure water to pure methanol, the electrification decreased (Figure 5B).
Towards a non-aqueous solution, we utilized diiodomethane with surface tension, $\gamma_{LV} =$
50.8 mN/m, and dielectric constant, $\epsilon_r \approx 5$ at normal temperature and pressure, which
could dissolve 1mM NaCl. We found that when droplets of diiodomethane (with or
without 1mM NaCl) are manually dispensed from polypropylene capillaries at the rate
of ~3 mL/min, they do not demonstrate measurable electrification. We consider this to be
due to the low dielectric constant, which prevents charge separation during the

dispensing of drops owing to the high electrostatic energy for charge separation, $U_E \propto$
$1/\epsilon_r$.

**Lines 310-316:** “Finally, we utilized diiodomethane (CH_2I_2) as a non-aqueous probe-liquid to
investigate its electrification when brought into contact with polypropylene. We dissolved 1 mM
NaCl in CH_2I_2 ; its surface tension, $\gamma \approx 51$ mN/m,¹⁴ facilitated the release of pendant droplets from
the polypropylene capillaries at a flow rate ~ 3 mL/min. The dispensed droplets (with or without
dissolved salt) did not have excess charge (Figure S7) because the low dielectric constant of
diiodomethane, $\epsilon_r = 5.3$,¹⁵ prevents charge separation during dispensing owing to the high
electrostatic energy, $U_E \propto 1/\epsilon_r$.¹⁶”

Q2.2 Later the authors state (line 360, pg 18), “The Debye length is the most important property
of aqueous solutions in the context of electrification at water-hydrophobe interfaces.” Given this
statement, it suggests that the authors should also study an additional electrolyte (eg KBr) that
does not influence the pH to demonstrate this is indeed the case. It is a reasonable explanation
but it would be good to establish this by additional experiments.

**A2.2 Thank you for this suggestion. Following the Reviewer’s comment, we performed new**
**experiments to measure the electrical charges carried by water droplets containing KBr in**
**the concentration range 10^{-7} - 10^1 M (Figure S6A) dispensed from polypropylene capillaries**
**at the rate of ~ 3 mL/min. The results of our experiments with KBr were quite similar to**
**those with NaCl (Figure S6). Also, please, note that all the salts utilized in this work were**
**baked at elevated temperatures to remove the organic impurities.**

 **Figure S6.** (A) Effects of ionic strength of KBr and NaCl solutions in the range 10^{-7} -1 M on the
 electrification at water-hydrophobe interfaces. The solutions were drawn into polypropylene
 pipettes from charge-neutral reservoirs and dispensed into a Faraday cage connected to an
 electrometer to quantify the electrical charge. (Dotted lines have been added to guide the eye). (B)
 Using an electrometer, we quantified the electrical charges of tiny aliquots from aqueous reservoirs
 whose ionic strength was adjusted by KBr, NaCl, NaOH, and HCl, using polypropylene pipettes.
 The electrical charges of the reservoirs were after the withdrawals⁴⁵ were always equal and
 opposite to those of the aliquots (blue). The magnitude of the electrification was quite similar for
 KBr (1mM), NaCl (1mM), and NaOH (pH 11), but it was significantly lower for HCl (pH 3). After
 the experiments with pH 3 solutions, when we used the same pipettes for water, the electrification
 corresponded to the original surface charge density of polypropylene. These results demonstrate
 that acids do not necessarily “neutralize” the surface charge. (Note: the surface charge densities
 were obtained by normalizing the observed charges by the solid-liquid interfacial areas inside the
 pipettes prior to dispensing.)

 Q2.3 The data supplied is more relevant to the effect of pH. As such more discussion of the
 effect of HCl and NaOH would be helpful. The data suggests that the polymer surface is nearly
 neutral at pH values below 4 and increases with pH, being fully charged beyond pH 7 – and at
 very high NaOH concentrations the Debye length becomes important (when the surface charge is
 maintained). This all fits with the influence of pH and ionic strength on the double layer.
 Similarly, I think the effect of CO₂ observed is best described by the effect it has on pH rather
 than the effect on the ionic strength or the species produced. The discussion regarding
 bicarbonate (lines 383-387) needs to be revisited. The presence of the bicarbonate ions resulting
 from CO₂ dissolution is irrelevant as they will be repelled by the negatively charged surface, it is
 the H⁺ liberated (as shown by the drop in pH) when CO₂ dissolves, which will negate the

negative surface charge at low pH. This is consistent with the data provided for HCl and
indicates that the effect of CO₂ can be understood as arising from its acidic properties.

**A2.3 We thank the reviewer for these terrific points.**

(i) To probe whether the acidic solution had “neutralized” the negatively charged
sites on the hydrophobic surface, as has been suggested before,^{7, 22, 42, 43, 44} we
performed new experiments to measure the absolute excess charge carried by
aqueous droplets from reservoirs of NaCl (1 mM), KBr (1 mM), NaOH (pH 11),
and HCl (pH 3). Crucially, we also measured the commensurate changes in the
charges of the reservoirs after pipetting. We found that the charges accrued in pH
3 reservoirs were significant lower in magnitude (albeit, equal and opposite to
those of the aliquots withdrawn) than those for the other three cases (Fig. S6B).
This means that rather than the H₃O⁺ ions getting permanently adsorbed to the
water-hydrophobe interface and “neutralizing” it, they were not even withdrawn
from the reservoir in the first place. Furthermore, the lack of the permanent
adsorption of the hydronium ions (or surface charge neutralization) is evident
from the fact that the electrification was completely recovered when the same tips
(as previously used for the pH 3 experiments) were used to pipette deionized
water. Now, the electrification corresponded to the original surface charge density
of the pipette (Figure S6). These data indicate that acids do not necessarily
“neutralize” the surface charge of the water-hydrophobe interface. While the
underlying mechanisms for this behavior of H₃O⁺ ions are beyond the scope of
this work, we surmise that the higher mobility and delocalization of protons in
water might facilitate higher electrical screening of the surface charge in
comparison to other cations.

**Changes made to the manuscript:**

**Lines 273-291:** “We found that by suppressing the Debye lengths by adding ions, the charge
carried by the pendant droplets dispensed from the hydrophobic capillaries decreased (Figure 5A;
see Figure S6A for the effects of varying ionic strength with KBr). Upon increasing the
concentration of HCl, the measured charge dropped significantly below pH 4, whereas for NaCl

and NaOH, the electrification was unaffected up until > 10 mM (Figure 5A). Indeed, ion-
 specificity has been reported on various interfacial and bulk properties^{32, 39, 40, 41}. To probe whether
 the acidic solution had “neutralized” the negatively charged sites on the hydrophobic surface, we
 measured the change in the charge of the reservoir as well. We found that the charges accrued in
 the acidic reservoirs were significantly lower in magnitude (albeit, equal and opposite to those of
 the aliquots withdrawn) than those for NaCl, KBr, and NaOH cases (Figure S6B). This means that
 rather than the H_3O^+ ions getting permanently adsorbed to the water-hydrophobe interface and
 “neutralizing” it, they were not even withdrawn from the reservoir in the first place. The lack of
 the permanent adsorption of the hydronium ions (or surface charge neutralization) is confirmed by
 the complete recovery of electrification when the same tips were used to pipette deionized water
 (Figure S6). These results prove that lower electrification with acids does not always mean that
 they “neutralize” the surface charge as previously believed.^{7, 22, 42, 43, 44} We surmise that the higher
 mobility of protons in water screens the electrical surface charge more effectively than hydrated
 cations.”

 **Figure S6.** (A) Effects of ionic strength of KBr and NaCl solutions in the range 10^{-7} -1 M on the
 electrification at water-hydrophobe interfaces. The solutions were drawn into polypropylene
 pipettes from charge-neutral reservoirs and dispensed into a Faraday cage connected to an
 electrometer to quantify the electrical charge. (Dotted lines have been added to guide the eye). (B)
 Using an electrometer, we quantified the electrical charges of tiny aliquots from aqueous reservoirs
 whose ionic strength was adjusted by KBr, NaCl, NaOH, and HCl, using polypropylene pipettes.
 The electrical charges of the reservoirs were after the withdrawals⁴⁵ were always equal and
 opposite to those of the aliquots (blue). The magnitude of the electrification was quite similar for
 KBr (1mM), NaCl (1mM), and NaOH (pH 11), but it was significantly lower for HCl (pH 3). After

the experiments with pH 3 solutions, when we used the same pipettes for water, the electrification
corresponded to the original surface charge density of polypropylene. These results demonstrate
that acids do not necessarily “neutralize” the surface charge. (Note: the surface charge densities
were obtained by normalizing the observed charges by the solid-liquid interfacial areas inside the
pipettes prior to dispensing.)

(ii) **We agree with the reviewer - indeed, the effect of CO₂ can be better explained by**
**the corresponding changes in the acidity of the solution. In the revised**
**manuscript, we explain it in lines 392-397:**

“We compared the electrification at water-polypropylene interfaces in two scenarios: water
supersaturated with CO₂ gas (pH ≈ 4), and water in equilibrium with the atmosphere (pH ≈ 5.6),
and found that the excess charges carried by the pendant water droplets decreased significantly
as the water became more acidic due to the formation of carbonic acid similarly to the HCl
solutions, as explained above (Figures S10, 5A and S6B).”

Q2.4 The clear suggestion from the data is that when a polymer surface is immersed in neat
methanol the surface charge is removed (we still expect a double layer in methanol, albeit of
reduced thickness). I believe that this is already known in the literature – (though I cannot find
the specific example at the moment – perhaps in the work of Galembeck). This suggests an
important additional experiment that could easily be performed without any additional materials
or change to the experimental set-up. By first removing the charge on the polymer (or
hydrophobic) surface using methanol the behavior of subsequent drops of water could be
measured to determine how the (positive) charges on the droplets change with time. This would
reveal the rate at which the negative charge on the polymer surface is regenerated whilst also
confirming the importance of this negative charge. Is the recovery very fast or slow? Does it
depend on humidity? Is it much quicker when immersed in water? I think such information will
be important in understanding the charging mechanism of hydrophobic surfaces which remains a
subject of great interest. Hence, I would really like to see this additional experiment added to the
paper.

**A2.4 We believe that the reviewer is referring to this paper by Galembeck⁴². In this work,**
**the co-authors (i) investigated the electrification of “general use” PTFE sheets by**

mechanically rubbing them against “general use” polyethylene foam, and (ii) tested the
efficacy of water, NaCl solutions, ethanol, *n*-hexane, and 1,1-difluoroethane at removing
those charges from the PTFE surface. The high surface charge density in these experiments
(~250 #/μm²) involved the transfer of mechanoradicals, followed by electron transfer,
depicted by the reaction scheme below (adapted from this work⁴²):

These hydrocarbocations and fluorocarbanions are shown to be mobile, forming positively
and negatively charged microscale “mosaics” on the surfaces. Next, they showed that
common polar and apolar solvents can specifically remove these (mobile) charged species
from the PTFE surfaces as follows: ethanol (both), *n*-hexane (cations), and water (poor at
removing both). They hypothesize that those differences might be due to the varying degrees
of solubility of the charged macromolecules in the solvents.

We would like to point out that the surface charge density of PTFE that we observe
in our experiments are about 25-times (2400%) lower than what we report in our study.
There is no rubbing involved, which leads to surface damage. In fact, we have explored this
class of electrification in the past: we electrified PTFE surfaces by rubbing them against
aluminum sheets. In these experiments (data presented in the Unpublishable SI file, Section
USI-2), we observed that the electrostatic charges on the rubbed PTFE surfaces were so high
(~10³ V/cm) that they could deflect pendant water droplets within ~12 mm distance. We also
observed that washing with acetone could effectively discharge the surfaces to their original
intrinsic charge level (data presented in the Unpublishable SI file, Section USI-2).

Following the Reviewer’s instruction, we performed the suggested experiments with
acetone as well as methanol. Using twenty brand new polypropylene pipettes, we measured
the electrical charges of the dispensed water droplets before and after washing them with
solvents, i.e. acetone and methanol, and drying overnight (Figure S9). We found that the
average charge of the dispensed water decreased by ~20% in either case after four cycles of
washing and drying.

Figure S9. Electrical charges of water droplets dispensed at the rate of ~ 3 mL/min from polypropylene pipettes and after washing them with (A) acetone or (B) methanol and drying overnight under ambient conditions. We also studied the effects of multiple washing cycles with a solvent, drying and dispensing water droplets four times for each pipette. The last set of bars represents the average results of performed measurements. The charges were measured by a Faraday cup connected to an electrometer.

**Changes made to the manuscript:**

**Lines 400-415:** “Lastly, Galembeck et al.,⁴² investigated the electrification at interfaces between
insulators (e.g., PTFE/polyethylene) as a function of rubbing speed, pressure, and duration. Under
these conditions, mechanoradicals form and they undergo electron transfers and self-assemble into
microscale “mosaics” comprising hydrocarbocations and fluorocarbanions (and with surface
charge densities > 25-times observed in our work). These mobile charges can be removed by
dissolving them into common polar and apolar solvents. To test whether the surface-bound charge
in our experiments can be removed by solvents, we utilized polypropylene pipettes and measured
the charges of the dispensed water droplets before and after washing them with acetone and
methanol (Figure S9). We found that the average charge of the dispensed water decreased by
~20% in either case, yet, the final surface charge density of the pipettes remained stable even after
four cycles of solvent washing and drying under our laboratory conditions. The 20% decrease in
the charge density may be due to the partial solubility of the polymer surface. These results
establish that the electrification that we observe is largely due to surface-bound immobile charges
and sub-surface charges that are inaccessible to solvents. Systematic investigations are needed to
unravel this.”

**Q2.5** Equation 3, why not replace L/V with $1/E$? Surely the electric field is the important
parameter and the same electric field could be achieved with different value of L provided the
L/V ratio was held constant.

**A2.5** Thank you for your comment, we have changed Eq. 3.

**Q2.6** Line 220 Should “diminish” be replaced with “change”? I suggest this as using the word
diminish does not rule out the charge density increasing.

**A2.6** Thank you for the suggestion, we have changed replaced the word “diminish”.

**Q2.7** The explanation on page 18 for why the droplets expressed from a liquid reservoir are
charged proportionally to the electric field is not completely clear to me. In this case there is a

net positive charge on the liquid and the electric field is described as promoting ion exchange
with the reservoir -presumably by repelling the positive charges (though isn't this direction
perpendicular to the electric field). Could the authors describe precisely how this leads to drops
that are negatively charged? One might think if the positive charge promotes exchange with the
reservoir this process would cease when then the positive charges have been neutralized.

**A2.7 Thank you for raising this simple but profound question.**

**In the revised manuscript, we provide a simple explanation for the observed ion-exchange**
**between the pendant drop and the liquid reservoir based on the electrical potential**
**difference. The electrical potential difference inside the capacitor ranges from V to zero as**
**one goes from the positively-charged plate to the grounded plate. This potential underlies the**
**ion-exchange: it pushes cations out of the pendant drop and pulls anions in; the higher the**
**electrical potential (or the electrical field within the capacitor, $E = V/L$), the larger the ion-**
**exchange, and hence the proportionality with the observed charge density.**

**Lines 365-372:** “For the air-filled capillary (Figure 2B), $\Delta q = 0$ because no charge can flow into
or out of the droplets. On the other hand, when the capillary is water-filled and/or connected to a
reservoir (Figure 2A and 2C), the higher electrical potential inside the capacitor drives ionic
current between the pendant droplet and the water reservoir and the ensuing charge $\Delta q \neq 0$ yields
the observed scaling behavior between the charge density with the applied field (Figure 3). Of
course, this ion-exchange sets up a potential difference along the length of the capillary, which is
why in the transitional case (Figure 2C), the initial droplets attract towards the positive plate, but
the latter ones repel away from it. Quantitative details of this process fall beyond the scope of this
work.”

We thank the reviewer for her/his thorough and constructive assessment of our manuscript. We
have responded to each reviewer comment (in red) below in **bold text**, and we have highlighted
all of the corresponding changes that we have made to the attached manuscript. We consider our
revised manuscript appropriate for publication in *Nature Communications*.

**Reviewer #3.**

**Comments:**

This manuscript presents experimental results aimed at understanding how different solid surfaces,
primarily hydrophobic, influence the charge of interfacial water. The results are clearly described
and thought provoking. I expect that these results will be quite interesting to the communities that
study aqueous liquid-solid interfaces. However, I think the manuscript falls short of providing
satisfying explanations for these results. Part of this is due to the rather ambitious claims made at
the end of the introduction about the scope of the questions that the manuscript claims to answer
(easily the topics of multiple separate studies), and part is due to the lack of consideration given to
alternate possible explanations for the observed behavior. I think these results should be published
but they belong under a more specialized journal title and without the use of overreaching overly
general conclusions.

**Response: Thank you for recognizing the importance of this contribution, which reports our**
**studies spanning over the last four years. In response to your constructive criticism, we have**
**made the following substantial changes:**

- 1. **Reduced the ambitious goals in the introduction, especially in the lines 62-63;**
**improved our explanations for the experimental results and conclusions.**
- 2. **Removed the section entitled “Previous Studies Revisited” which critiqued on**
**alternate possible explanations;**
- 3. **We have thoroughly investigated various other possible explanations for the**
**electrification. Please, see**
 - a. **Lines 209-221 (streaming currents)**
 - b. **Lines 278-289 (adsorption of hydroxide ions)**
 - c. **Lines 378-382 (ionizable functional groups)**
 - 747 d. **Lines 382-284 (surface-bound electrons)**
 - e. **Lines 391-399 (adsorption of bicarbonate ions)**

- f. **Lines 400-415** (mechanoradicals)
- 750 g. **Lines 457-461** (dipolar arrangement of interfacial water and the partial charge
- transfer between interfacial water and the hydrophobe)
- 4. In the revised manuscript package, we have added new experimental results on
- a. the effects of flow-speed on the electrification (**lines 209-221**),
- b. the effects of dielectric constants on the electrification by using diiodomethane
- and diiodomethane containing 1 mM NaCl as probe-liquids (**lines 310-316**),
- c. quantifying the magnitude of charge transfer when pH 3 solutions are
- withdrawn, proving that acids do not necessarily “neutralize” the surface
- charge, as previously believed^{7, 22, 42, 43, 44} (**lines 273-291**), and
- 759 d. cleaning hydrophobic surfaces with organics (acetone and methanol) to test if
- the surface-bound charges can be removed, as has been noted previously⁴²
- (**lines 400-415**).
- 5. We also attach as an **Unpublishable SI file** reporting some new experimental results
- on the release of ions from the electrical double layer in the absence of external
- electrical field (**Section USI-1**), which might interest the Reviewer.

With these substantial changes to the manuscript and a thorough revision to enhance its

lucidity, we sincerely hope that the Reviewer finds the revised manuscript worthy of

publication in *Nature Communications*.

**Regarding the suitability of this work for *Nature Communications*, we wish to bring to the**

**Editor’s attention the point-of-views of the other two Reviewers:**

- (i) Reviwer#1: “I think the experimental approach is quite original and clearly shows that water
- droplet electrification is ion-based and depends on wetting-dewetting conditions.”
- (ii) Reviwer#2: “This article deals with a phenomenon that has been known for more than a
- century but remains poorly understood. This attests to the challenge in unravelling the
- details of the electrification of water in contact with hydrophobic surfaces. One of the
- complications has been the difficulty in obtaining reliable reproducible experiments that can
- be used to evaluate various explanations of the effect. I find the conceptually simple
- experiments described here to be a very important contribution to the field as they clearly

- demonstrate that
- 1) The hydrophobic surfaces have a negative charge (even in air)
 - 2) This results in water acquiring a positive charge when it is imbibed into the capillary
 - 3) This is then manifest in positively charged droplets when the water is released from the
 - capillaries
 - 4) This effect is observed when the capillary is not filled with liquid

I think the clever experimental design and the quality of the results make these arguments
very easy to prosecute and as such I expect that this work will be highly impactful in a
number of fields from colloid science to industrial processing.”

Q3.1 The principle conclusion is that hydrophobic surfaces carry an excess negative surface charge
and that this charge promotes the accumulation of positive charge at the water interface. I find this
explanation to be plausible, however, the extent to which this finding is general to hydrophobic
surfaces or pertains to just the surface under investigation here is not adequately resolved.

**A3.1 Thank you. We agree with the Reviewer’s point. In the revised manuscript, we have**
**added the following statement to the conclusion:**

**Lines 438-439: “We should specify that the conclusions drawn in this work about polypropylene,**
**PTFE, and FDTS might not translate to all hydrophobic materials.”**

Q3.2 The authors assume that hydrophobicity of the capillary ensures that no residual water film
remains after the drop is ejected. Further testing should be done to confirm this point. Coulomb
interactions are strong and I expect that if the interior of the capillary does indeed feature surface
bound electrons, then those regions would be effectively hydrophilic and potentially strong enough
to promote the sticking of the excess H_3O^+ .

**A3.2 Thank you. The Reviewer is correct in pointing out that tiny micro/nano droplets might**
**be left behind inside the hydrophobic capillary. In the revised version, we list this as a**
**potential source of experimental error. However, it should be realized that as water is drawn**
**inside a hydrophobic capillary from an electroneutral reservoir, it brings in excess positive**

ions and the reservoir acquires a negative charge (Figure 4A-C). The fact that those two
charges are equal and opposite proves that the residual water on the capillary surface is not
a significant source of error in our experiments or their interpretation.

**Changes to the manuscript:**

**Lines 423-427:** “Hydrophobicity ensures that when water droplets are dispensed from capillaries,
they do not leave thin film behind, like in the case of hydrophilic surfaces.^{30, 38} Although nano-
and microdroplets might remain on surface defects,¹⁷ presumably, containing a tiny fraction of
the excess charge, most of the water exits the pipette along with the bulk of the excess charge.”

**Lines 445-448:** “The unavoidable hydrolysis and cross-linking of silanes, manual dispensing,
variations in the ambient conditions, left-over nano/micro droplets inside pipettes, and the
polarization of capillaries inside the capacitor could contribute to experimental errors.”

**Q3.3** Much of the interest and debate over this topic in the field has focused on unraveling the
relative thermodynamic preference of H_3O^+ or OH^- to populate hydrophobic or air-water
interfaces. By positing hydrophobic surface charge effects, insight into the native preference of
OH^- or H_3O^+ for interfacial solvation is not advanced in this manuscript.

**A3.3 Thank you for this important comment. We address it in three points:**

(i) The focus of this manuscript is to bring forward a hitherto unrealized source of
the electrification of solid hydrophobes due to their innate surface charge. Thus,
the water-air interface is not the primary focus of this work.

(ii) For the water-hydrophobe interfaces investigated here, the specific adsorption of
H_3O^+ or OH^- ions has negligible significance. We demonstrate it by the following:

a. In the **Supplementary Section S3 and Table S1**, we prove that the
electrification at the air-water interface is negligible in comparison to that at
the interfaces of water with polypropylene, FDTS, and PTFE.

b. In **Lines 278-289**, we demonstrate that in acidic solutions, the electrification is
weaker and it is not due to the titration of surface-bound OH^- ions, as
previously believed^{7, 22, 42, 43, 44}. New experimental results added (Fig. S6B)

**Changes made to the manuscript:**

**Lines 203-204:** “In addition, we found that the electrification of water droplets in our experiments
was negligibly influenced by the air-water interfacial area (Section S3, Table S1).”

**Lines 273-291:** “We found that by suppressing the Debye lengths by adding ions, the charge
carried by the pendant droplets dispensed from the hydrophobic capillaries decreased (Figure 5A;
see Figure S6A for the effects of varying ionic strength with KBr). Upon increasing the
concentration of HCl, the measured charge dropped significantly below pH 4, whereas for NaCl
and NaOH, the electrification was unaffected up until > 10 mM (Figure 5A). Indeed, ion-
specificity has been reported on various interfacial and bulk properties^{32, 39, 40, 41}. To probe whether
the acidic solution had “neutralized” the negatively charged sites on the hydrophobic surface, we
measured the change in the charge of the reservoir as well. We found that the charges accrued in
the acidic reservoirs were significantly lower in magnitude (albeit, equal and opposite to those of
the aliquots withdrawn) than those for NaCl, KBr, and NaOH cases (Figure S6B). This means that
rather than the H₃O⁺ ions getting permanently adsorbed to the water-hydrophobe interface and
“neutralizing” it, they were not even withdrawn from the reservoir in the first place. The lack of
the permanent adsorption of the hydronium ions (or surface charge neutralization) is confirmed by
the complete recovery of electrification when the same tips were used to pipette deionized water
(Figure S6). These results prove that lower electrification with acids does not always mean that
they “neutralize” the surface charge as previously believed.^{7, 22, 42, 43, 44} We surmise that the higher
mobility of protons in water screens the electrical surface charge more effectively than hydrated
cations.”

 **Figure S6.** (A) Effects of ionic strength of KBr and NaCl solutions in the range 10^{-7} -1 M on the
 electrification at water-hydrophobe interfaces. The solutions were drawn into polypropylene
 pipettes from charge-neutral reservoirs and dispensed into a Faraday cage connected to an
 electrometer to quantify the electrical charge. (Dotted lines have been added to guide the eye). (B)
 Using an electrometer, we quantified the electrical charges of tiny aliquots from aqueous reservoirs
 whose ionic strength was adjusted by KBr, NaCl, NaOH, and HCl, using polypropylene pipettes.
 The electrical charges of the reservoirs after the withdrawals⁴⁵ were always equal and
 opposite to those of the aliquots (blue). The magnitude of the electrification was quite similar for
 KBr (1mM), NaCl (1mM), and NaOH (pH 11), but it was significantly lower for HCl (pH 3). After
 the experiments with pH 3 solutions, when we used the same pipettes for water, the electrification
 corresponded to the original surface charge density of polypropylene. These results demonstrate
 that acids do not necessarily “neutralize” the surface charge. (Note: the surface charge densities
 were obtained by normalizing the observed charges by the solid-liquid interfacial areas inside the
 pipettes prior to dispensing.)

 Q3.4 Furthermore, I find that the accumulation of negative charge in droplets in contact with
 hydrophilic surfaces, or for hydrophobic surfaces including a capillary water reservoir are not
 adequately explained.

 A3.4 Thank you. In the revised manuscript, we provide a simple explanation for the observed
 ion-exchange between the pendant drop and the liquid reservoir based on the electrical
 potential difference. The electrical potential difference inside the capacitor ranges from V to
 zero as one goes from the positively-charged plate to the grounded plate. This potential
 underlies the ion-exchange: it pushes cations out of the pendant drop and pulls anions in; the

**higher the electrical potential (or the electrical field within the capacitor, $E = V/L$), the larger**
**the ion-exchange, and hence the proportionality with the observed charge density.**

**Lines 365-372: “On the other hand, when the capillary is water-filled and/or connected to a**
**reservoir (Figure 2A and 2C), the higher electrical potential inside the capacitor drives ionic**
**current between the pendant droplet and the water reservoir and the ensuing charge $\Delta q \neq 0$ yields**
**the observed scaling behavior between the charge density with the applied field (Figure 3). Of**
**course, this ion-exchange sets up a potential difference along the length of the capillary, which is**
**why in the transitional case (Figure 2C), the initial droplets attract towards the positive plate, but**
**the latter ones repel away from it. Quantitative details of this process fall beyond the scope of this**
**work.”**

**Q3.5 I’m not sure data set (i) in Fig. 3B is convincingly constant across increasing field strengths.**
**Since constant behavior is central to the conclusions of the manuscript, there would be benefit to**
**working towards reducing the error bars in this data.**

**A3.5 Thank you for your comments. We have revisited the experiments, added new points**
**for the air-filled and water-filled cases and modified Figure 3. We have also improved the**
**aesthetics of the figure.**

**Figure 3.** Experiments inside a parallel plate capacitor. Correlations between the charge density
 of pendant drops and electric field strengths for water-filled and air-filled hydrophobic and
 hydrophilic capillaries (Movies S4-10). **Red** arrows in the insets present qualitatively the direction
 and the relative magnitudes of the deflections experienced by water drops under various scenarios.
 When the hydrophobic capillaries were air-filled (or water-filled), the pendant water drops repelled
 away from (or attracted towards) the positively charged plate. In the case of the hydrophilic glass
 capillaries, the droplets always tilted towards the positive plate of the capacitor, and we detected
 very low charges in the air-filled case, and higher charges in (iv) the filled case. Note: we found
 no deflection for droplets of methanol and hexadecane dispensed from air-filled hydrophobic
 capillaries in the specified range of electric fields. We obtained the charge density along the y-axis
 by normalizing the charge of the droplet obtained from the force balance [eq.(1-4)] and liquid-
 solid interfacial areas of the dispensed drops. (Dotted lines have been added to guide the eye.)

**Q3.6** In Fig. 4C there appears to be systematic shift in the charge of dispensed drops
 consecutively over the first 100 drops. Why would this be?

**A3.6** Thank you. Electrification experiments are notorious for variability, as has been noted
 by several research groups.^{37, 46} In these experiments, the drops were manually dispensed by
 the lead author; the experiments could be sensitive to the leftover microdroplets during

**dispensing; cleaning of the surface of the pipette tip; ambient contamination; static**
**electricity of the experimentalist, etc. It is hard to pinpoint the exact reasons, unfortunately.**
**That is why we performed 300 sequential experiments and reported the mean and the**
**standard deviation. These experiments clearly show that the pipetted water carries an extra**
**positive charge, from which we deduced that the surface is negatively charged. We also**
**submit that the absolute value is not the most crucial factor here because it would vary for**
**other materials. In the revised manuscript, we list some of the possible sources of error:**

**Lines 445-448: “The unavoidable hydrolysis and cross-linking of silanes, manual dispensing,**
**variations in the ambient conditions, left-over nano/micro droplets inside pipettes, and the**
**polarization of capillaries inside the capacitor could contribute to experimental errors.”**

**References:**

- 1. Vemulapalli GK, Kukolich SG. Why does a stream of water deflect in an electric field?
*Journal of Chemical Education* **73**, 887-888 (1996).
- 2. Strazdaite S, Versluis J, Backus EH, Bakker HJ. Enhanced ordering of water at
hydrophobic surfaces. *The Journal of Chemical Physics* **140**, 054711 (2014).
- 3. Matyushov DV. Electrophoretic mobility without charge driven by polarisation of the
nanoparticle–water interface. *Molecular Physics* **112**, 2029-2039 (2014).
- 4. Vácha R, *et al.* The Orientation and Charge of Water at the Hydrophobic Oil Droplet–
Water Interface. *Journal of the American Chemical Society* **133**, 10204-10210 (2011).
- 5. Vacha R, Marsalek O, Willard AP, Bonthuis DJ, Netz RR, Jungwirth P. Charge Transfer
between Water Molecules As the Possible Origin of the Observed Charging at the
Surface of Pure Water. *Journal of Physical Chemistry Letters* **3**, 107-111 (2012).
- 6. Poli E, Jong KH, Hassanali A. Charge transfer as a ubiquitous mechanism in determining
the negative charge at hydrophobic interfaces. *Nature Communications* **11**, 901 (2020).
- 7. Beattie JK. The intrinsic charge on hydrophobic microfluidic substrates. *Lab Chip* **6**,
1409-1411 (2006).
- 8. Liu C, Bard AJ. Electrostatic electrochemistry at insulators. *Nature Materials* **7**, 505
(2008).
- 9. Liu CY, Bard AJ. Chemical Redox Reactions Induced by Cryptoelectrons on a PMMA
Surface. *Journal of the American Chemical Society* **131**, 6397-6401 (2009).
- 10. Li S, *et al.* Contributions of Different Functional Groups to Contact Electrification of
Polymers. *Advanced Materials* **n/a**, 2001307 (2020).
- 11. Faubel M, Steiner B. Strong Bipolar Electrokinetic Charging of Thin Liquid Jets
Emerging from 10 μm PtIr Nozzles. *Berichte der Bunsengesellschaft für physikalische*
*Chemie* **96**, 1167-1172 (1992).
- 12. Duffin AM, Saykally RJ. Electrokinetic Hydrogen Generation from Liquid Water
Microjets. *The Journal of Physical Chemistry C* **111**, 12031-12037 (2007).
- 13. Schwierz N, *et al.* Hydrogen and Electric Power Generation from Liquid Microjets:
Design Principles for Optimizing Conversion Efficiency. *Journal of Physical Chemistry*
*C* **120**, 14513-14521 (2016).

- 14. Strom G, Fredriksson M, Stenius P. Contact Angles, Work of Adhesion, and Interfacial-
Tensions at a Dissolving Hydrocarbon Surface. *Journal of Colloid and Interface Science*
**119**, 352-361 (1987).
- 15. Lide DR. *CRC Handbook of Chemistry and Physics, 84th Edition*. Taylor & Francis
(2003).
- 16. Feynman RP, Leighton RB, Sands M. *The Feynman Lectures on Physics (Vol. II)*.
Addison-Wesley Publishing Company Inc. and Narosa Publishing House (2003).
- 17. Wong WSY, *et al.* Microdroplet Contaminants: When and Why Superamphiphobic
Surfaces Are Not Self-Cleaning. *ACS Nano* **14**, 3836-3846 (2020).
- 18. Marx V. Pouring over liquid handling. *Nature Methods* **11**, 33 (2013).
- 19. Choi D, *et al.* Spontaneous electrical charging of droplets by conventional pipetting.
*Scientific Reports* **3**, (2013).
- 20. Choi D, Tsao Y-H, Chiu C-M, Yoo D, Lin Z-H, Kim DS. A smart pipet tip:
Triboelectricity and thermoelectricity assisted in situ evaluation of electrolyte
concentration. *Nano Energy* **38**, 419-427 (2017).
- 21. Wu C, Wang AC, Ding W, Guo H, Wang ZL. Triboelectric Nanogenerator: A
Foundation of the Energy for the New Era. *Advanced Energy Materials* **9**, 1802906
(2019).
- 22. McCarty LS, Whitesides GM. Electrostatic charging due to separation of ions at
interfaces: Contact electrification of ionic electrets. *Angewandte Chemie-International*
*Edition* **47**, 2188-2207 (2008).
- 23. Xu W, *et al.* A droplet-based electricity generator with high instantaneous power density.
*Nature* **578**, 392-396 (2020).
- 24. Sarver T, Al-Qaraghuli A, Kazmerski LL. A comprehensive review of the impact of dust
on the use of solar energy: History, investigations, results, literature, and mitigation
approaches. *Renewable & Sustainable Energy Reviews* **22**, 698-733 (2013).
- 25. Pratt TH. *Electrostatic Ignitions of Fires and Explosions*. John Wiley & Sons (2010).
- 26. Wilkinson M. Large deviation analysis of rapid onset of rain showers. *Physical Review*
*Letters* **116**, 018501 (2016).
- 27. Shavlov AV, Zdzhumandzhi VA. Metastable states and coalescence of charged water
drops inside clouds and fog. *Journal of Aerosol Science* **91**, 54-61 (2016).

- 28. Schlumpberger S, Lu NB, Suss ME, Bazant MZ. Scalable and Continuous Water
Deionization by Shock Electrodialysis. *Environmental Science & Technology Letters* **2**,
367-372 (2015).
- 29. Sun Q, *et al.* Surface charge printing for programmed droplet transport. *Nature Materials*,
(2019).
- 30. Domingues EM, Arunachalam S, Nauruzbayeva J, Mishra H. Biomimetic coating-free
surfaces for long-term entrapment of air under wetting liquids. *Nature Communications*
**9**, 3606 (2018).
- 31. Das R, Ahmad Z, Nauruzbayeva J, Mishra H. Biomimetic Coating-free
Superomniphobicity. *Scientific Reports* **10**, 7934 (2020).
- 32. Mazzini V, Craig VSJ. What is the fundamental ion-specific series for anions and
cations? Ion specificity in standard partial molar volumes of electrolytes and
electrostriction in water and non-aqueous solvents. *Chemical Science* **8**, 7052-7065
(2017).
- 33. McCaffrey DL, *et al.* Mechanism of ion adsorption to aqueous interfaces:
Graphene/water vs. air/water. *Proceedings of the National Academy of Sciences* **114**,
13369 (2017).
- 34. Keithley Instruments I. Low Level Measurement Handbook. Precision DC Current,
Voltage, and Resistance Measurements (Edition 6). (ed[^](eds) (2004).
- 35. *CRC Handbook of Chemistry and Physics*. CRC Press (1999).
- 36. Lowell J, Truscott WS. Triboelectrification of Identical Insulators .2. Theory and Further
Experiments. *Journal of Physics D-Applied Physics* **19**, 1281-1298 (1986).
- 37. Lowell J, Akande AR. Contact electrification-why is it variable? *Journal of Physics D:*
*Applied Physics* **21**, 125-137 (1988).
- 38. Mishra H, *et al.* Time-Dependent Wetting Behavior of PDMS Surfaces with Bioinspired,
Hierarchical Structures. *ACS Applied Materials & Interfaces* **8**, 8168-8174 (2016).
- 39. Mishra H, Enami S, Nielsen RJ, Hoffmann MR, Goddard WA, Colussi AJ. Anions
dramatically enhance proton transfer through aqueous interfaces. *P Natl Acad Sci USA*
**109**, 10228-10232 (2012).
- 40. Enami S, Mishra H, Hoffmann MR, Colussi AJ. Hofmeister effects in micromolar
electrolyte solutions. *The Journal of Chemical Physics* **136**, 154707 (2012).
- 41. Beattie JK, *et al.* pH and the surface tension of water. *Journal of Colloid and Interface*
*Science* **422**, 54-57 (2014).

42. Burgo TAL, Ducati TRD, Francisco KR, Clinckspoor KJ, Galembeck F, Galembeck SE.
Triboelectricity: Macroscopic Charge Patterns Formed by Self-Arraying Ions on Polymer
Surfaces. *Langmuir* **28**, 7407-7416 (2012).
43. Healy T, Fuerstenau D. The isoelectric point/point-of zero-charge of interfaces formed by
aqueous solutions and nonpolar solids, liquids, and gases. *Journal of colloid and*
*interface science* **309**, 183-188 (2007).
44. Karraker KA, Radke CJ. Disjoining pressures, zeta potentials and surface tensions of
aqueous non-ionic surfactant/electrolyte solutions: theory and comparison to experiment.
*Advances in Colloid and Interface Science* **96**, 231-264 (2002).
45. Subramanian N, *et al.* Evaluating the potential of superhydrophobic nanoporous alumina
membranes for direct contact membrane distillation. *Journal of Colloid and Interface*
*Science* **533**, 723-732 (2019).
46. Lacks DJ, Mohan Sankaran R. Contact electrification of insulating materials. *Journal of*
*Physics D: Applied Physics* **44**, 453001 (2011).

REVIEWERS' COMMENTS:

Reviewer #1 (Remarks to the Author):

Very nice! The paper has very much gained and now reads very smoothly and the findings are nicely supported by a logic chain of arguments.

The questions raised by me earlier are adequately addressed or commented.

Introducing the caveats section now clearly draws the line between presented findings and open points.

I recommend to publish this nice article with the option of a small revision at the authors discretion considering the following minor points:

Minor points:

polypropylen surface charge quantification without mention of charging mechanism somewhat unclear at this early position in the text - rephrase with a larger context?

check format of long explicite reference listings; e.g. 3-8 instead of 3, 4, 5, 6, 7, 8,

how is droplet angle measured (shape analysis? error?), how is droplet shape affected by charge distribution in the droplet?

is the falling trajectory angle concurrent with the charge deduced from pending angle α ?

Fig 1c potential of drop dispenser not shown (grounded?)

Discussion: to help the reader you should elaborate on how you deduce that the PP surface must be negatively charged even in air?

How does your experiment exclude hydrophobe surface electrification upon initial contact with liquid?

Manfred Heuberger

Reviewer #2 (Remarks to the Author):

Second review of “Electrification at water-hydrophobe surfaces”, Nauruzbayeva et al

I am satisfied with the extensive extra work and responses that the authors have provided (both to my comments and others) and recommend that the manuscript is accepted for publication. In particular the manuscript is now more clearly written and the arguments prosecuted more clearly. In making this recommendation I note that this work raises as many questions as it answers. This is not a bad thing. I feel the work represents a significant and important advance to the field that will enable researchers to newly identify exciting new avenues of research across a wide area of research activities, therefore it warrants publication in this prestigious journal.

Comments the authors may wish to consider

The authors report the surface charge density of the hydrophobic polypropylene surface to be -0.7 nC cm^{-2} . This can be compared to the surface charge density obtained from fitting the DLVO theory to surface force measurements. This fitting yields the surface potential which can be converted to the surface charge density using the Grahame equation. In figure 4 of Meagher and Craig[1], the surface forces have been measured between two hydrophobic (advancing angle >100 degrees, receding angle 90 degrees) polypropylene surfaces in aqueous solutions. A surface potential of 38 mV is obtained at a concentration of $1.4 \times 10^{-4} \text{ M NaCl}$ and the surface charge density for the latter is $1.11 \times 10^{-3} \text{ C m}^{-2}$ or 111 nC cm^{-2} . This much higher surface charge is due to dissociation in aqueous solution. It demonstrates that only about 1 in 1000 possible ionisation sites (most likely surface hydroxyl or carboxyl groups) need remain charged in air to be consistent with the measurements reported here. This highlights the sensitivity of the measurements reported here and that the actual charge densities obtained are very small. Further, an increase in humidity should lead to an increase in adsorption of water at the surface, which would favour further ionisation, consistent with the observed effect of humidity.

The above is also highly relevant to the section entitled “ Why do water-hydrophobe interfaces become electrically charged”. For polymer surfaces the answer is simply that because they exist in an oxidising environment there exist surface groups that can ionise when placed in a high dielectric medium – the existence of these groups is not incommensurate with the surface being hydrophobic (eg PP above but also silanated silica is hydrophobic but also highly charged see [2]). The real mystery is why some of these charges are retained in air, even at near zero humidity and this is

currently open to speculation. The evidence suggests that surface damage through rubbing increases their number, so the effect of topography or low levels of surface contamination should not be ruled out.

Regarding Lines 272-291

Here the authors have responded to my suggestion that the effect of acid is to titrate the charged groups. I very much like the additional experiment reported where they show that this can be traced back to imbibition where they see that no charge is left on the reservoir when an acid is used. I assume the results reported are the mean and std dev of a number of measurements. If the acid in solution is neutralising the surface charge (and this effect remains when the solution is removed) this may well occur on the very first imbibition of acid solution into the capillary, after which the charge on the surface is neutralised and remaining measurements show no charge transfer. Note contact angle experiments show that the contact angles of surfaces with pH sensitive groups can be altered by pre-soaking the surface in solution and then drying the surface before measurement - showing that the neutralising/ionising effect of solution can remain when the liquid is removed.

When the solution is changed for water the effective change in pH would see the surface charge renewed, consistent with the observations. So I would suggest that the surface charge titration explanation cannot be ruled out by these measurements unless it is very clear that no charge is transferred on the very first occasion that the pipette is exposed to the acid solution (or the first occasion after it has been rinsed with water and the charge reestablished). Perhaps this can be checked before this mechanism is ruled out?

Lines 289-291

“The higher mobility of protons screens the electrical surface charge more effectively than hydrated cations?”

To me this statement is not sustainable. The mobility of the cations and the protons should enable equilibration on a timescale that is very much faster than the experiments, so the difference in mobility should not lead to an observable influence.

Sincerely,

Vincent S. J. Craig

References

[1] L. Meagher, V.S.J. Craig, Effect of Dissolved Gas and Salt on the Hydrophobic Force between Polypropylene Surfaces, *Langmuir* 10 (1994) 2736-2742.

[2] T.D. Blake, J.A. Kitchener, Stability of Aqueous Films on Hydrophobic Methylated Silica, *J. Chem. Soc, Faraday Trans. I* 68 (1972) 1435-1442.

Reviewer #3 (Remarks to the Author):

The revised version of this manuscript has significantly improved in focus and clarity. The experiments are well designed and reveal clear physical insight into the systems being studied. My remaining concern with the manuscript is that I think the generality of the conclusions with regard to hydrophobic surfaces are still overstated.

Electrification has been observed in a wide range of circumstances, including apparently at the water-air interface. The mechanism uncovered here, while very interesting, seems to be specific to the kinds of surfaces considered being studied here (i.e., polypropylene, FDTS, and PTFE). The results convincingly show that these materials carry negative surface charge and that the interactions of this charge with H_3O^+ explains the observed charging of droplets. The corollary seems to be that this mechanism underlies surface charging that has been observed in other contexts, which I'm skeptical about. Hydrophobicity is key to these observations because it allows droplets to be cleanly ejected, however, it does not follow that all hydrophobic surfaces carry negative charge. In fact, it would seem possible that positively charged hydrophobic surfaces exhibit the opposite charging (via the same mechanism) and that neutral hydrophobic surfaces exhibit no charging at all. I think it's important that the authors either properly contextualize their findings in this way, or provide more convincing evidence that hydrophobic materials somehow imply negative surface charge.

Thus, my perception is that this very nice work is of limited generality to hydrophobic surface charging and thus more appropriate for a more specialized journal. This is partly because it is not clear to me that these specific materials are sufficiently prevalent (or that the origins of their surface charging tendencies are widely problematic) to warrant publication in Nature Communication. However, I would leave it those more familiar with the potential applications of these surfaces to evaluate whether the potential broader impacts of these materials overcome my reservations.

We thank the reviewer (Prof. Manfred Heuberger) for his thorough and constructive assessment of our manuscript. We have responded to each comment (in red) below in **bold text**, and we have listed the changes that we have made in the revised manuscript. We consider our revised manuscript to be appropriate for publication in *Nature Communications*.

(**Note:** The line numbers for the changes in the manuscript correspond to the “No Markup” version of the manuscript. We opted for this because when changes are tracked in a .docx file, line numbers can become discontinuous. For a better reading experience, we are also uploading the revised manuscript without “track changes” as a Supporting File.)

Reviewer #1.

Comments:

Very nice! The paper has very much gained and now reads very smoothly and the findings are nicely supported by a logic chain of arguments. The questions raised by me earlier are adequately addressed or commented. Introducing the caveats section now clearly draws the line between presented findings and open points. I recommend to publish this nice article with the option of a small revision at the authors discretion considering the following minor points.

Response: We sincerely thank the Reviewer for pushing us to produce a substantially improved version of the manuscript.

Q 1.1 Polypropylene surface charge quantification without mention of charging mechanism somewhat unclear at this early position in the text - rephrase with a larger context?

A 1.1 Thank you. We are assuming that the Reviewer is referring to the abstract. In the revised version, we have removed the polypropylene surface charge from the abstract and broadened the context of our findings.

Changes to the manuscript: Lines 27-30: “Based on these results, we deduced that common hydrophobic materials possess surface-bound negative charge. Thus, when these surfaces are submerged in water, hydrated cations form an electrical double layer.”

Q 1.2 check format of long explicite reference listings; e.g. 3-8 instead of 3, 4, 5, 6, 7, 8.

A 1.2 Thank you. The Editor has assured us that the *Nature Communications*' Production Team will take care of reference formatting.

Q 1.3 how is droplet angle measured (shape analysis? error?), how is droplet shape affected by charge distribution in the droplet? is the falling trajectory angle concurrent with the charge deduced from pending angle alpha?

A 1.3 Thank you for these three questions, which we address point-wise.

Shape analysis: we recorded movies of deflections/tilting of pendant droplets inside a parallel plate capacitor as a function of the electrical voltage applied to the positively charged plate. In order to quantify the tilting angles, the frames of the movies were imported into the AutoCAD software (Step 1, Figure A1.3.1). For each tilting measurement, we zoomed into the image and selected several pixels at the liquid-vapor interface and utilized the Spline command to create a smooth curve through those points, thereby realizing a 2D closed region (Step 2, Figure A1.3.1). Next, the geometric center (C_x , C_y) was determined using the MASSPROP command (Step 3, Figure A1.3.1)¹ by dividing the first moment of the area with the area:

$$A = \iint dx dy \quad [1]$$

$$C_x = \frac{1}{A} \iint x dx dy ; C_y = \frac{1}{A} \iint y dx dy \quad [2]$$

Next, a line is drawn connecting the center of mass of the droplet and the geometric center of the tip of the pipette where the pendant drop is formed. Assuming the droplet to be axisymmetric about this line, we realized that the geometric center and the 3D center of mass overlap. Therefore, we consider the point (C_x , C_y) to be the center of mass of the droplet. The tilting angles (α) are thus measured between this axis and a vertical line passing through the center of the capillary (Step 4, Figure A1.3.1). Potential sources of errors in this analysis include, non-axis-symmetric droplets, contact angle hysteresis at the solid-liquid-vapor interface, and low resolution of images during spline-fitting².

Figure A1.3.1. Droplet shape analysis procedure for a representative frame extracted from a supplementary movie. Step 1: The frame is imported into the AutoCAD software. Step 2: 2D region is defined by choosing points at the liquid-vapor interface and joining them via Spline functions. Step 3: The geometric center is estimated by dividing the first moment of the area by the area, using the MASSPROP command [Eqs. 1-2]. Step 4: Tilting angle (α) is measured between the vertical line passing through the capillary's center and the line joining the center of the capillary's tip and the center of mass.

Changes to the manuscript: Lines 385-388: “Additionally, this analysis does not account for the effects of the contact angle hysteresis at the capillary-water-air interface, the evaporation of water, and the non-uniform distribution of the excess charge in the droplets' bulk and at the air-water interface.”

Charge distribution inside droplets: This is an important but complex question. In our estimations of the electrical force, we assume the excess charge to be located (as a lump) at the center of mass of the droplet. This simplifies the consideration of forces due to the electric and gravitational fields. In reality, however, the excess charge is distributed both inside the drop and at the air-water interface.³ In fact, the fractionation of solvated ions in water is an important unresolved problem in chemical science,^{4,5} and, perhaps, the fractionation might be non-monotonic under the effects of an electric field.⁶ Thus, we are not in a position to address this question in detail at this point.

Changes to the manuscript: Lines 385-388: “Additionally, this analysis does not account for the effects of the contact angle hysteresis at the capillary-water-air interface, the evaporation of water, and the non-uniform distribution of the excess charge in the droplets’ bulk and at the air-water interface.”

Droplet trajectory: In our capacitor experiments, the pendant droplets experience two orthogonal forces, namely: the gravitational force (mg , downwards) and the electrical force (qE , horizontally). While the gravitational force stays the same, we keep increasing the electric field until the droplet detaches from the capillary. Correspondingly, the tilting angles at various applied voltages are representative of the resultant force, $\tan \alpha = F_E/F_G$, and this allows us to calculate the excess charge inside the droplet with reasonable accuracy. Eventually, the droplet detaches from the capillary and continues in a straight line, i.e., along the vector sum of the electrical and gravitational forces. In Figure A1.3.2, we present snapshots of droplets for the water-filled case (top panel) and the air-filled case (bottom panel) just before and after detaching from the capillary. These snapshots, derived from Supplementary Movies 1 and 2, demonstrate that the droplets indeed move along straight lines defined by the tilting angles at the time of the detachment.

Changes to the manuscript: Lines 136-139: “The tilting angles increased with the electric field strength, eventually leading to the detachment of the droplet from the capillary above; the trajectories followed straight lines along the vector sum of the forces due to gravity and electrostatics (Supplementary Movies 1-2).”

Figure A1.3.2. Trajectories of pendant droplets after detachment from the capillary under the action of orthogonal gravitational and electrical fields. (A) The water-filled case: the droplet travels along a straight line defined by the tilting angle at the time of detachment (frames extracted from Supplementary Movie 1). (B) The air-filled case: the droplet travels along a straight line defined by the tilting angle at the time of detachment (frames extracted from Supplementary Movie 2).

Q 1.4 Fig 1c potential of drop dispenser not shown (grounded?)

A 1.4 The “dispenser” refers to the capillary that is comprised of polypropylene, PTFE, or FDTS-coated glass. We have modified the figure and labelled the dispenser as “capillary” to avoid any confusion.

Q 1.5 Discussion: to help the reader you should elaborate on how you deduce that the PP surface must be negatively charged even in air?

A 1.5 Thank you. This point has been elaborated in the following sections of the manuscript.

Lines 236-254: “To investigate the origins of water electrification we considered whether: (i) the excess charge originated at the interface of the water and the hydrophobic surface due to, for instance, the selective adsorption/desorption of H_3O^+ or OH^- ; or (ii) if the hydrophobic capillaries selectively drew water with excess charge during the intake process. To test these hypotheses, we placed a water reservoir inside a Faraday cage and monitored changes in its electrical charge during water withdrawal/addition. Specifically, we placed 1 mL of water (reservoir) inside the Faraday cage and we extracted aliquots (15-50 μL) using capillaries of varying wettability, while logging the charge of the water reservoir (Figure 4A). Simultaneously, we added these extracted aliquots to another electrometer and measured the response. We found that when we extracted water using hydrophobic capillaries made of polypropylene, FDTS-coated glass, and polytetrafluoroethylene tubes, the charge on the bulk water reservoir became negative (Figure 4B, red bars). Interestingly, when we added these extracted water aliquots to another electrometer using the same hydrophobic capillaries, we recovered equal and opposite positive charges (Figure 4B, blue bars). From these data we deduced that polypropylene, FDTS-coated glass, and polytetrafluoroethylene surfaces have surface charge densities of $\sigma = -0.7 \pm 0.1 \text{ nC cm}^{-2}$ (or $43 \pm 7 \# \mu\text{m}^{-2}$), $-0.46 \pm 0.11 \text{ nC cm}^{-2}$ (or $29 \pm 7 \# \mu\text{m}^{-2}$), and $-0.12 \pm 0.04 \text{ nC cm}^{-2}$ (or $7 \pm 2 \# \mu\text{m}^{-2}$), respectively. From the chemical composition of these materials, we also know that they do not have Brønsted acid groups.”

Lines 339-346: “From our experimental results, we have deduced that the surfaces of solid hydrophobes such as polypropylene, or polytetrafluoroethylene (PTFE) or FDTS-coated glass are negatively charged even in air. When water comes into contact with these surfaces, the ions present in the water, such as H_3O^+ , OH^- and other cations/anions, form an electrical double layer with excess positive charges, and this charge separation is facilitated by the water’s high dielectric constant⁸. Thus, when a hydrophobic capillary draws water in from a reservoir, it draws in excess positive charges and leaves behind an equal and opposite negative excess charge at the source (Figures 4, 6).”

Lines 464-468: “Our experimental results demonstrate that when water comes into contact with common hydrophobic materials such as polypropylene, FDTS, and PTFE, cations partition to the solid-liquid interface. Thus, these surfaces have a negative surface charge density. The electrical double layer theory predicts ensuring trends: liquids with lower dielectric constants and higher ionic strengths experience lower electrification.”

Q 1.6 How does your experiment exclude hydrophobe surface electrification upon initial contact with liquid?

A 1.6 Thank you for this question. If the electrification manifests upon the initial contact with the liquid, it will entail the production of positive and negative ions, such that the system will be electrically neutral overall. This might be followed by the specific adsorption of ions at the water-hydrophobe interface, as has been proposed by others, e.g, Beattie & co-workers⁹⁻¹⁵. However,

- i) through Figures 2B & 3 (the air-filled cases), we prove that the water inside the capillary carries an excess of positive charge only, i.e., it is not electroneutral;**
- ii) in Figure 4A-B, we quantify this positive charge and show that the reservoir accrues an equal and opposite negative charge; the simple pipetting experiment – drawing in positively charged water and dispensing it – is quite reproducible as shown in 300 measurements in Figure 4C. Based on these data, we deduce that hydrophobic capillaries must carry a net negative surface charge. This is the main message of this paper.**
- iii) And, in SI Section S3, we prove that the air-water interface is an insignificant contributor, if at all, to the excess charge carried by drops pipetted from hydrophobic capillaries.**

The above-mentioned points have been discussed in the manuscript in the following sections:

Lines 49-58: “A variety of mechanisms have been put forth to explain electrification of water in contact with solid/liquid/gaseous hydrophobes, including the specific adsorption of hydroxide ions^{10-14,16-21} and hydronium ions²²⁻²⁹, the dipolar organization of interfacial water³⁰⁻³², the partial charge transfer between the O and H atoms of interfacial water^{33,34} or between interfacial water and oil molecules³⁵, the adsorption of bicarbonate ions due to the dissolution of ambient CO₂³⁶, contamination^{15,37-41}, reactive chemical groups⁴²⁻⁴⁶, electrons trapped on the surface of insulators⁴⁷⁻⁴⁹ and mechanoradicals^{50,51}. With the exception of surface-bound electrons, these mechanisms assume that common hydrophobic surfaces such as polypropylene and perfluorocarbons are electrically neutral in air.”

Lines 135-141: “..when the capillary was air-filled, the droplets were repelled away from the positively charged plate (Figures 2B and 3, and Supplementary Movie 2). The tilting angles increased with the electric field strength, eventually leading to the detachment of the droplet from the capillary above; the trajectories followed straight lines along the vector sum of the forces due to gravity and electrostatics (Supplementary Movies 1-2). In the air-filled scenario,

the estimated charge densities of droplets dispensed from hydrophobic and hydrophilic capillaries remained unaffected by the electric fields (Figure 3).”

Lines 236-254: “To investigate the origins of water electrification we considered whether: (i) the excess charge originated at the interface of the water and the hydrophobic surface due to, for instance, the selective adsorption/desorption of H_3O^+ or OH^- ; or (ii) if the hydrophobic capillaries selectively drew water with excess charge during the intake process. To test these hypotheses, we placed a water reservoir inside a Faraday cage and monitored changes in its electrical charge during water withdrawal/addition. Specifically, we placed 1 mL of water (reservoir) inside the Faraday cage and we extracted aliquots (15-50 μL) using capillaries of varying wettability, while logging the charge of the water reservoir (Figure 4A). Simultaneously, we added these extracted aliquots to another electrometer and measured the response. We found that when we extracted water using hydrophobic capillaries made of polypropylene, FDTS-coated glass, and polytetrafluoroethylene tubes, the charge on the bulk water reservoir became negative (Figure 4B, red bars). Interestingly, when we added these extracted water aliquots to another electrometer using the same hydrophobic capillaries, we recovered equal and opposite positive charges (Figure 4B, blue bars). From these data we deduced that polypropylene, FDTS-coated glass, and polytetrafluoroethylene surfaces have surface charge densities of $\sigma = -0.7 \pm 0.1 \text{ nC cm}^{-2}$ (or $43 \pm 7 \# \mu\text{m}^{-2}$), $-0.46 \pm 0.11 \text{ nC cm}^{-2}$ (or $29 \pm 7 \# \mu\text{m}^{-2}$), and $-0.12 \pm 0.04 \text{ nC cm}^{-2}$ (or $7 \pm 2 \# \mu\text{m}^{-2}$), respectively. From the chemical composition of these materials, we also know that they do not have Brønsted acid groups.”

Lines 203-216: “Faraday cup measurements (Figure 1C, Methods, Supplementary Section 2, and Supplementary Figure 3) demonstrate that the excess positive charges on the water droplets dispensed from hydrophobic capillaries were proportional to the solid-liquid interfacial area. For instance, for the conical polypropylene capillaries, the excess electrical charges carried by five 10 μL droplets were greater than those carried by 50 μL droplets due to its conical geometry (Supplementary Figure 4A). However, for the cylindrical FDTS-coated glass capillaries, the charges carried by five 10 μL droplets were equal to the charges carried by 50 μL droplets (Supplementary Figure 4B). In addition, we found that the electrification of water droplets in our experiments was negligibly influenced by the air-water interfacial area (Supplementary Section 3, Supplementary Table 1). Furthermore, if the same polypropylene tip was used to consecutively dispense 300 droplets of 50 μL volume from a large water reservoir into the Faraday cup connected to the electrometer, the net charge or charge density of those droplets did not change significantly over time (Figure 4C).”

We thank the reviewer (Professor Vincent Craig) for his thorough and constructive assessment of our manuscript. We have responded to each reviewer comment (in red) below in **bold text**, and we have tracked all of the corresponding changes that we have made to the attached manuscript. We consider our revised manuscript appropriate for publication in *Nature Communications*.

(**Note:** Line numbers in the response below correspond to the “No Markup” version of the revised manuscript file. We did this because when changes are tracked in a .docx file, line numbers become discontinuous. For a better reading experience, we are also uploading the revised manuscript without “track changes” as a Supporting File.)

Reviewer #2.

Comments:

I am satisfied with the extensive extra work and responses that the authors have provided (both to my comments and others) and recommend that the manuscript is accepted for publication. In particular the manuscript is now more clearly written and the arguments prosecuted more clearly. In making this recommendation I note that this work raises as many questions as it answers. This is not a bad thing. I feel the work represents a significant and important advance to the field that will enable researchers to newly identify exciting new avenues of research across a wide area of research activities, therefore it warrants publication in this prestigious journal.

Response: We are grateful to the Reviewer for recognizing the significance of our work and helping us in substantially improving its quality.

Q 2.1 The authors report the surface charge density of the hydrophobic polypropylene surface to be -0.7 nC cm^{-2} . This can be compared to the surface charge density obtained from fitting the DLVO theory to surface force measurements. This fitting yields the surface potential which can be converted to the surface charge density using the Grahame equation. In figure 4 of Meagher and Craig[1], the surface forces have been measured between two hydrophobic (advancing angle >100 degrees, receding angle 90 degrees) polypropylene surfaces in aqueous solutions. A surface potential of 38 mV is obtained at a concentration of $1.4 \times 10^{-4} \text{ M NaCl}$ and the surface charge density for the latter is $1.11 \times 10^{-3} \text{ C m}^{-2}$ or 111 nC cm^{-2} . This much higher surface charge is

due to dissociation in aqueous solution. It demonstrates that only about 1 in 1000 possible ionisation sites (most likely surface hydroxyl or carboxyl groups) need remain charged in air to be consistent with the measurements reported here. This highlights the sensitivity of the measurements reported here and that the actual charge densities obtained are very small. Further, an increase in humidity should lead to an increase in adsorption of water at the surface, which would favour further ionisation, consistent with the observed effect of humidity.

A 2.1 Thank you for bringing this insightful and relevant article to our attention. Indeed, the electrostatic repulsion measured between polypropylene surfaces by AFM⁵² translated to a dramatically higher surface charge density than what is reported in our manuscript. We do not know how to reconcile those two measurements. Surface preparation and handling, surface roughness, and environmental factors could certainly impact electrification; though, (hydrophobic) confinement of water in the 30-100 nm range should not lead to dramatic consequences, as noted by Meagher & Craig as well⁵². We, however, do not think that our observations could be explained entirely based on the presence of Brønsted acid groups on polypropylene, FDTS, and polytetrafluoroethylene. It is because

- (i) such groups would get re-protonated as water would dewet the surface, and
- (ii) if thin films of water persisted on the surface, containing the EDL, then the electrification should (monotonically) decrease if the same pipette pip is used over and over to dispense water; but that is not the case (Fig. 4C).

These points have been discussed in the manuscript and listed below as well.

Changes in the manuscript: Lines 449-454: “Additionally, it is worth noting that AFM-based measurements of electrostatic repulsion between polypropylene surfaces in water at sub-30 nm separations reveal a significantly higher surface charge density ($\sim 111 \text{ nC/cm}^2$)⁵² than ours. Further investigation is thus needed to systematically unravel the contributions of surface roughness, surface preparation and handling, and the ambient atmosphere in these experiments.”

Also, please see Lines 459-462: “The unavoidable hydrolysis and cross-linking of silanes, manual dispensing, variations in the ambient conditions, left-over nano/micro droplets inside pipettes, and the polarization of capillaries inside the capacitor could contribute to experimental errors.”

Lines 396-401: “On the other hand, dielectrics with ionizable functional groups, such as silica⁵³, get deprotonated/protonated depending on the pH- pK_a , thereby creating an electrical double layer⁵⁴. However, when water leaves the surface, those chemical groups are

reprotonated/deprotonated to become electrically neutral. Thus, pendant drops pipetted from capillaries with Brønsted acid/base groups present non-significant charging (Figure 4B).”

Q 2.2 The above is also highly relevant to the section entitled “ Why do water-hydrophobe interfaces become electrically charged”. For polymer surfaces the answer is simply that because they exist in an oxidising environment there exist surface groups that can ionise when placed in a high dielectric medium – the existence of these groups is not incommensurate with the surface being hydrophobic (eg PP above but also silanated silica is hydrophobic but also highly charged see [2]). The real mystery is why some of these charges are retained in air, even at near zero humidity and this is currently open to speculation. The evidence suggests that surface damage through rubbing increases their number, so the effect of topography or low levels of surface contamination should not be ruled out.

A 2.2 Thank you. We have addressed this point in A2.1 above. Please, note that we have removed the sub-headings in the Discussion section, following the editorial guidelines.

Q 2.3 Regarding Lines 272-291

Here the authors have responded to my suggestion that the effect of acid is to titrate the charged groups. I very much like the additional experiment reported where they show that this can be traced back to imbibition where they see that no charge is left on the reservoir when an acid is used. I assume the results reported are the mean and std dev of a number of measurements. If the acid in solution is neutralising the surface charge (and this effect remains when the solution is removed) this may well occur on the very first imbibition of acid solution into the capillary, after which the charge on the surface is neutralised and remaining measurements show no charge transfer. Note contact angle experiments show that the contact angles of surfaces with pH sensitive groups can be altered by pre-soaking the surface in solution and then drying the surface before measurement -showing that the neutralising/ionising effect of solution can remain when the liquid is removed.

When the solution is changed for water the effective change in pH would see the surface charge renewed, consistent with the observations. So I would suggest that the surface charge titration explanation cannot be ruled out by these measurements unless it is very clear that no charge is transferred on the very first occasion that the pipette is exposed to the acid solution (or the first

occasion after it has been rinsed with water and the charge reestablished). Perhaps this can be checked before this mechanism is ruled out?

A 2.3 Thank you for seeking clarity in the experiments that you had originally proposed.

1. The red and blue bars and the error bars in the Supplementary Figure 6B refer to the mean and standard deviation of at least ten consecutive measurements, respectively. First, we presented excess electrical charges accrued inside a pH 3 reservoir as tiny droplets are withdrawn from it. Next, we used this pipette to remove droplets from DI water bulk reservoir several times.
2. Following the Reviewer's question, we used a polypropylene pipette tip to measure the excess charge carried by a single droplet of pH3 water (Fig. A2.3). Then, we used this pipette tip to draw deionized (DI) water drop and measure its excess charge; DI water drops were then drawn five more times and their charges were measured. This experiment (pH 3 → DI water → DI water five times) was repeated three times with three different pipette tips and we present these data in Fig. A2.3. We found that the charge densities of the DI water droplets drawn right after the tip's exposure to pH 3 water were significantly higher; the observed charge density of dispensed DI water did not change significantly on the subsequent drawings. Lastly, the charge densities observed for the DI water were comparable to those presented in a similar experiment reported in Supplementary Figure 6B.

Figure A2.3. Surface charge density of a pH 3 water droplet (green) withdrawn using a polypropylene pipette tip. Surface charge density of a DI water drop withdrawn right after

the tip's exposure to pH 3 water (blue). If the same tip is used to draw water five more times, the average value is the similar to that observed during the first drawing of DI water. This experiment was repeated three times with new pipettes and it gave consistent results.

Q 2.4 Lines 289-291: “The higher mobility of protons screens the electrical surface charge more effectively than hydrated cations?”

To me this statement is not sustainable. The mobility of the cations and the protons should enable equilibration on a timescale that is very much faster than the experiments, so the difference in mobility should not lead to an observable influence.

A 2.4 Thank you, we have deleted that statement.

References

- [1] L. Meagher, V.S.J. Craig, Effect of Dissolved Gas and Salt on the Hydrophobic Force between Polypropylene Surfaces, *Langmuir* 10 (1994) 2736-2742.
- [2] T.D. Blake, J.A. Kitchener, Stability of Aqueous Films on Hydrophobic Methylated Silica, *J. Chem. Soc, Faraday Trans. I* 68 (1972) 1435-1442.

We thank the reviewer for her/his thorough and constructive assessment of our manuscript. We have responded to each reviewer comment (in red) below in **bold text**, and we have tracked all of the corresponding changes that we have made to the attached manuscript. We consider our revised manuscript appropriate for publication in *Nature Communications*.

(**Note:** Line numbers in the response below correspond to the “No Markup” version of the revised manuscript file. We did this because when changes are tracked in a .docx file, line numbers become discontinuous. For a better reading experience, we are also uploading the revised manuscript without “track changes” as a Supporting File.)

Reviewer #3.

Comments:

The revised version of this manuscript has significantly improved in focus and clarity. The experiments are well designed and reveal clear physical insight into the systems being studied. My remaining concern with the manuscript is that I think the generality of the conclusions with regard to hydrophobic surfaces are still overstated.

Response: Thank you very much.

Q 3.1 Electrification has been observed in a wide range of circumstances, including apparently at the water-air interface. The mechanism uncovered here, while very interesting, seems to be specific to the kinds of surfaces considered being studied here (i.e., polypropylene, FDTS, and PTFE). The results convincingly show that these materials carry negative surface charge and that the interactions of this charge with H_3O^+ explains the observed charging of droplets. The corollary seems to be that this mechanism underlies surface charging that has been observed in other contexts, which I'm skeptical about. Hydrophobicity is key to these observations because it allows droplets to be cleanly ejected, however, it does not follow that all hydrophobic surfaces carry negative charge. In fact, it would seem possible that positively charged hydrophobic surfaces exhibit the opposite charging (via the same mechanism) and that neutral hydrophobic surfaces exhibit no charging at all. I think it's important that the authors either properly

contextualize their findings in this way, or provide more convincing evidence that hydrophobic materials somehow imply negative surface charge.

Thus, my perception is that this very nice work is of limited generality to hydrophobic surface charging and thus more appropriate for a more specialized journal. This is partly because it is not clear to me that these specific materials are sufficiently prevalent (or that the origins of their surface charging tendencies are widely problematic) to warrant publication in Nature Communication. However, I would leave it those more familiar with the potential applications of these surfaces to evaluate whether the potential broader impacts of these materials overcome my reservations.

A 3.1 Thank you for these comments.

- (i) This report focuses on aqueous interfaces with solid hydrophobic materials; thus, we have not focused on the air-water interface. However, we have demonstrated that the electrification at the air-water interface does not significantly influence our findings (Supplementary Section 3).**
- (ii) We thank the Reviewer for questioning the generality of these findings and bringing up the corollary regarding reverse electrification, if a hydrophobic surface has intrinsic positive charge. Yes, our findings do not apply to all hydrophobic materials; and, yes, we expect the electrification trends to reverse for positively-charged surfaces. These points appear in the Discussion section, and we list them below:**

Lines 448-449: “We should specify that the conclusions drawn in this work about polypropylene, PTFE, and FDTS might not translate to all hydrophobic materials.”

Lines 477-478: “In principle, the electrification trends could reverse, e.g., from negative to positive, for a hydrophobic surface with positive charge.”

References

- Wilson, J. *3D Modeling in AutoCAD: Creating and Using 3D Models in AutoCAD 2000, 2000i, 2002, and 2004*. (CRC Press, 2001).
- Liu, K., Vuckovac, M., Latikka, M., Huhtamäki, T. & Ras, R. H. A. Improving surface-wetting characterization. *Science* **363**, 1147, doi:10.1126/science.aav5388 (2019).
- McCaffrey, D. L. *et al.* Mechanism of ion adsorption to aqueous interfaces: Graphene/water vs. air/water. *Proceedings of the National Academy of Sciences* **114**, 13369, doi:10.1073/pnas.1702760114 (2017).
- Onsager, L. & Samaras, N. N. T. The Surface Tension of Debye-Huckel Electrolytes. *Journal of Chemical Physics* **2**, doi:10.1063/1.1749522 (1934).
- Craig, V. S. J., Cui, J. & Brazier, T. G. in *Aqua Incognita: Why Ice Floats on Water and Galileo 400 Years on* (ed P Lonostro and B W Ninham) 341-349 (Connor Court Publishing, 2014).
- Baskin, A. & Prendergast, D. Improving Continuum Models to Define Practical Limits for Molecular Models of Electrified Interfaces. *Journal of The Electrochemical Society* **164**, E3438-E3447, doi:10.1149/2.0461711jes (2017).
- Keithley Instruments, I. Low Level Measurement Handbook. Precision DC Current, Voltage, and Resistance Measurements (Edition 6). (2004).
- Feynman, R. P., Leighton, R. B. & Sands, M. *The Feynman Lectures on Physics (Vol. II)*. (Addison-Wesley Publishing Company Inc. and Narosa Publishing House, 2003).
- Beattie, J. K. & Djerdjev, A. M. The Pristine Oil/Water Interface: Surfactant-Free Hydroxide-Charged Emulsions. **43**, 3568-3571, doi:10.1002/anie.200453916 (2004).
- Beattie, J. K. The intrinsic charge on hydrophobic microfluidic substrates. *Lab Chip* **6**, 1409-1411, doi:10.1039/b610537h (2006).
- Beattie, J. K. Comment on Autoionization at the surface of neat water: is the top layer pH neutral, basic, or acidic? by R. Vacha, V. Buch, A. Milet, J. P. Devlin and P. Jungwirth, *Phys. Chem. Chem. Phys.*, 2007, 9, 4736. *Phys. Chem. Chem. Phys.* **10**, 330-331, doi:10.1039/b713702h (2008).
- Beattie, J. K., Djerdjev, A. M. & Warr, G. G. The surface of neat water is basic. **61**, 31--39, doi:10.1039/b805266b (2009).
- Creux, P., Lachaise, J., Graciaa, A., Beattie, J. K. & Djerdjev, A. M. Strong Specific Hydroxide Ion Binding at the Pristine Oil/Water and Air/Water Interfaces. *J. Phys. Chem. B* **113**, 14146-14150, doi:10.1021/jp906978v (2009).
- Gray-Weale, A. & Beattie, J. K. An explanation for the charge on water's surface. *Phys. Chem. Chem. Phys.* **11**, 10994-11005, doi:10.1039/b901806a (2009).
- Beattie, J. K. & Gray-Weale, A. Oil/Water Interface Charged by Hydroxide Ions and Deprotonated Fatty Acids: A Comment. *Angewandte Chemie International Edition* **51**, 12941-12942, doi:doi:10.1002/anie.201205927 (2012).
- Karraker, K. A. & Radke, C. J. Disjoining pressures, zeta potentials and surface tensions of aqueous non-ionic surfactant/electrolyte solutions: theory and comparison to experiment. *Advances in Colloid and Interface Science* **96**, 231-264, doi:[https://doi.org/10.1016/S0001-8686\(01\)00083-5](https://doi.org/10.1016/S0001-8686(01)00083-5) (2002).
- Zangi, R. & Engberts, J. B. F. N. Physisorption of hydroxide ions from aqueous solution to a hydrophobic surface. *Journal of the American Chemical Society* **127**, 2272-2276, doi:10.1021/ja044426f (2005).

- Mundy, C. J., Kuo, I. F. W., Tuckerman, M. E., Lee, H. S. & Tobias, D. J. Hydroxide anion at the air-water interface. *Chem Phys Lett* **481**, 2-8, doi:DOI 10.1016/j.cplett.2009.09.003 (2009).
- Tian, C. S. & Shen, Y. R. Structure and charging of hydrophobic material/water interfaces studied by phase-sensitive sum-frequency vibrational spectroscopy. *Proceedings of the National Academy of Sciences of the United States of America* **106**, 15148-15153, doi:10.1073/pnas.0901480106 (2009).
- Marinova, K. G. *et al.* Charging of Oil–Water Interfaces Due to Spontaneous Adsorption of Hydroxyl Ions. *Langmuir* **12**, 2045-2051, doi:10.1021/la950928i (1996).
- Healy, T. & Fuerstenau, D. The isoelectric point/point-of zero-charge of interfaces formed by aqueous solutions and nonpolar solids, liquids, and gases. *Journal of colloid and interface science* **309**, 183-188, doi:10.1016/j.jcis.2007.01.048 (2007).
- Petersen, M. K., Iyengar, S. S., Day, T. J. F. & Voth, G. A. The hydrated proton at the water liquid/vapor interface. *J. Phys. Chem. B* **108**, 14804-14806, doi:10.1021/jp046716o (2004).
- Iuchi, S., Chen, H. N., Paesani, F. & Voth, G. A. Hydrated Excess Proton at Water-Hydrophobic Interfaces. *J Phys Chem B* **113**, 4017-4030, doi:Doi 10.1021/Jp805304j (2009).
- Kathmann, S. M., Kuo, I. F. W., Mundy, C. J. & Schenter, G. K. Understanding the Surface Potential of Water. *Journal of Physical Chemistry B* **115**, 4369-4377, doi:Doi 10.1021/Jp1116036 (2011).
- Petersen, P. B. & Saykally, R. J. Evidence for an enhanced hydronium concentration at the liquid water surface. *J. Phys. Chem. B* **109**, 7976-7980, doi:10.1021/jp044479j (2005).
- Zhang, C. *et al.* Water at hydrophobic interfaces delays proton surface-to-bulk transfer and provides a pathway for lateral proton diffusion. *Proceedings of the National Academy of Sciences of the United States of America* **109**, 9744-9749, doi:Doi 10.1073/Pnas.1121227109 (2012).
- Yamaguchi, S., Kundu, A., Sen, P. & Tahara, T. Communication: Quantitative estimate of the water surface pH using heterodyne-detected electronic sum frequency generation. *Journal of Chemical Physics* **137**, doi:Artn 151101 10.1063/1.4758805 (2012).
- Rizzuto, A. M., Cheng, E. S., Lam, R. K. & Saykally, R. J. Surprising Effects of Hydrochloric Acid on the Water Evaporation Coefficient Observed by Raman Thermometry. *Journal of Physical Chemistry C* **121**, 4420-4425, doi:10.1021/acs.jpcc.6b12851 (2017).
- Schwierz, N. *et al.* Hydrogen and Electric Power Generation from Liquid Microjets: Design Principles for Optimizing Conversion Efficiency. *Journal of Physical Chemistry C* **120**, 14513-14521, doi:10.1021/acs.jpcc.6b03788 (2016).
- Vemulapalli, G. K. & Kukulich, S. G. Why does a stream of water deflect in an electric field? *Journal of Chemical Education* **73**, 887-888 (1996).
- Matyushov, D. V. Electrophoretic mobility without charge driven by polarisation of the nanoparticle–water interface. *Molecular Physics* **112**, 2029-2039, doi:10.1080/00268976.2014.882521 (2014).
- Smolentsev, N. & Roke, S. Self-assembly at water nanodroplet interfaces quantified with nonlinear light scattering. *Langmuir*, doi:10.1021/acs.langmuir.0c01887 (2020).

- Vácha, R. *et al.* The Orientation and Charge of Water at the Hydrophobic Oil Droplet–Water Interface. *Journal of the American Chemical Society* **133**, 10204-10210, doi:10.1021/ja202081x (2011).
- Vacha, R. *et al.* Charge Transfer between Water Molecules As the Possible Origin of the Observed Charging at the Surface of Pure Water. *Journal of Physical Chemistry Letters* **3**, 107-111 (2012).
- Poli, E., Jong, K. H. & Hassanali, A. Charge transfer as a ubiquitous mechanism in determining the negative charge at hydrophobic interfaces. *Nature Communications* **11**, 901, doi:10.1038/s41467-020-14659-5 (2020).
- Yan, X. B. *et al.* Central Role of Bicarbonate Anions in Charging Water/Hydrophobic Interfaces. *Journal of Physical Chemistry Letters* **9**, 96-103, doi:10.1021/acs.jpcclett.7b02993 (2018).
- Harper, W. R. Liquids giving no electrification by bubbling. *British Journal of Applied Physics* **4**, S19-S22, doi:10.1088/0508-3443/4/s2/308 (1953).
- Roger, K. & Cabane, B. Why Are Hydrophobic/Water Interfaces Negatively Charged? *Angewandte Chemie-International Edition* **51**, 5625-5628, doi:10.1002/anie.201108228 (2012).
- Jena, K. C., Scheu, R. & Roke, S. Surface Impurities Are Not Responsible For the Charge on the Oil/Water Interface: A Comment. *Angewandte Chemie International Edition* **51**, 12938-12940, doi:doi:10.1002/anie.201204662 (2012).
- Roger, K. & Cabane, B. Uncontaminated Hydrophobic/Water Interfaces Are Uncharged: A Reply. *Angewandte Chemie International Edition* **51**, 12943-12945, doi:10.1002/anie.201207114 (2012).
- Uematsu, Y., Bonthuis, D. J. & Netz, R. R. Charged Surface-Active Impurities at Nanomolar Concentration Induce Jones-Ray Effect. *Journal of Physical Chemistry Letters* **9**, 189-193, doi:10.1021/acs.jpcclett.7b02960 (2018).
- Choi, D. *et al.* Spontaneous electrical charging of droplets by conventional pipetting. *Scientific Reports* **3**, doi:ARTN 2037 10.1038/srep02037 (2013).
- Cole, J. J., Barry, C. R., Knuesel, R. J., Wang, X. & Jacobs, H. O. Nanocontact electrification: Patterned surface charges affecting adhesion, transfer, and printing. *Langmuir* **27**, 7321--7329, doi:10.1021/la200773x (2011).
- Diaz, A. F. & Felix-Navarro, R. M. A semi-quantitative tribo-electric series for polymeric materials: the influence of chemical structure and properties. *Journal of Electrostatics* **62**, 277-290, doi:<https://doi.org/10.1016/j.elstat.2004.05.005> (2004).
- Horn, R. G., Smith, D. T. & Grabbe, A. Contact electrification induced by monolayer modification of a surface and relation to acid–base interactions. *Nature* **366**, 442-443, doi:10.1038/366442a0 (1993).
- Choi, D. *et al.* A smart pipet tip: Triboelectricity and thermoelectricity assisted in situ evaluation of electrolyte concentration. *Nano Energy* **38**, 419-427, doi:<https://doi.org/10.1016/j.nanoen.2017.06.020> (2017).
- Lowell, J. & Truscott, W. S. Triboelectrification of Identical Insulators .2. Theory and Further Experiments. *Journal of Physics D-Applied Physics* **19**, 1281-1298, doi:Doi 10.1088/0022-3727/19/7/018 (1986).
- Liu, C. & Bard, A. J. Electrostatic electrochemistry at insulators. *Nature Materials* **7**, 505, doi:10.1038/nmat2160

<https://www.nature.com/articles/nmat2160#supplementary-information> (2008).

- Liu, C. Y. & Bard, A. J. Chemical Redox Reactions Induced by Cryptoelectrons on a PMMA Surface. *Journal of the American Chemical Society* **131**, 6397-6401, doi:10.1021/ja806785x (2009).
- Baytekin, B., Baytekin, H. T. & Grzybowski, B. A. What Really Drives Chemical Reactions on Contact Charged Surfaces? *Journal of the American Chemical Society* **134**, 7223-7226, doi:10.1021/ja300925h (2012).
- Piperno, S., Cohen, H., Bendikov, T., Lahav, M. & Lubomirsky, I. The Absence of Redox Reactions for Palladium(II) and Copper(II) on Electrostatically Charged Teflon: Relevance to the Concept of “Cryptoelectrons”. *Angewandte Chemie International Edition* **50**, 5654-5657, doi:10.1002/anie.201101203 (2011).
- Meagher, L. & Craig, V. S. J. Effect of Dissolved Gas and Salt on the Hydrophobic Force between Polypropylene Surfaces. *Langmuir* **10**, 2736-2742, doi:10.1021/la00020a039 (1994).
- Vigil, G., Xu, Z. H., Steinberg, S. & Israelachvili, J. Interactions of Silica Surfaces. *Journal of Colloid and Interface Science* **165**, 367-385, doi:DOI 10.1006/jcis.1994.1242 (1994).
- Yutkin, M. P., Mishra, H., Patzek, T. W., Lee, J. & Radke, C. J. Bulk and Surface Aqueous Speciation of Calcite: Implications for Low-Salinity Waterflooding of Carbonate Reservoirs. *SPE Journal* **23**, 84-101, doi:10.2118/182829-PA (2018).